

# Simulating secondary organic aerosol from missing diesel-related intermediate-volatility organic compound emissions during the Clean Air for London (ClearfLo) campaign

R. Ots[1,2], D. E. Young[3,*], M. Vieno[2], L. Xu[4], R. E. Dunmore[5], J. D. Allan[3,6], H. Coe[3], L. R. Williams[7], S. C. Herndon[7], N. L. Ng[4,8], J. F. Hamilton[5], R. Bergström[9,10], C. Di Marco[2], E. Nemitz[2], I. A. Mackenzie[11], J. J. P. Kuenen[12], D. C. Green[13], S. Reis[2,14], and M. R. Heal[1]

[1]School of Chemistry, University of Edinburgh, Edinburgh, UK
[2]Natural Environment Research Council, Centre for Ecology & Hydrology, Penicuik, UK
[3]School of Earth, Atmospheric and Environmental Sciences, University of Manchester, Manchester, UK
[4]School of Chemical and Biomolecular Engineering, Georgia Institute of Technology, Atlanta, GA, USA
[5]Wolfson Atmospheric Chemistry Laboratories, Department of Chemistry, University of York, York, UK
[6]National Centre for Atmospheric Science, University of Manchester, Manchester, UK
[7]Aerodyne Research, Inc., Billerica, MA, USA
[8]School of Earth and Atmospheric Sciences, Georgia Institute of Technology, Atlanta, GA, USA
[9]Swedish Meteorological and Hydrological Institute, Norrköping, Sweden
[10]Department of Chemistry and Molecular Biology, University of Gothenburg, Gothenburg, Sweden
[11]School of GeoSciences, University of Edinburgh, Edinburgh, UK
[12]TNO, Department of Climate, Air and Sustainability, Utrecht, The Netherlands
[13]MRC PHE Centre for Environment and Health, King's College London, London, UK
[14]University of Exeter Medical School, Knowledge Spa, Truro, UK
[*]now at: Department of Environmental Toxicology, University of California, Davis, CA, USA

*Correspondence to:* M. Heal (M.Heal@ed.ac.uk) and R. Ots (riinu.ots@gmail.com)

**Abstract.**

We present high-resolution atmospheric chemistry transport model (ACTM) simulations of secondary organic aerosol (SOA) formation over the UK for 2012. Our simulations include additional diesel-related intermediate volatility organic compound (IVOC) emissions derived directly from comprehensive field measurements at an urban background site in London during the

5    2012 Clean Air for London (ClearfLo) campaign. Our IVOC emissions are added proportionally to VOC emissions, as opposed to proportionally to primary organic aerosol (POA) as has been done by previous ACTM studies seeking to simulate the effects of these missing emissions. Modelled concentrations are evaluated against hourly and daily measurements of organic aerosol (OA) components derived from aerosol mass spectrometer (AMS) measurements also made during the ClearfLo campaign at three sites in the London area. Good hourly performance in comparison to the measurements was shown, giving confidence in

10    the SOA prediction skill of the ACTM system used.

According to the model simulations, diesel-related IVOCs can explain on average ~30% of the annual SOA in and around London. Furthermore, the 90-th percentile of modelled daily SOA concentrations for the whole year is 3.8 µg m$^{-3}$ (more than 40% of which is produced from the missing diesel precursors), constituting a notable addition to total particulate matter. More measurements of these precursors (currently not included in official emissions inventories) is recommended.





During the period of concurrent measurements, SOA concentrations at the Detling rural background location east of London were greater than at the central London location. The model shows that this was caused by an intense pollution plume with a strong gradient of imported SOA passing over the rural location. This demonstrates the value of modelling for supporting the interpretation of measurements taken at different sites or for short durations.

# 1  Introduction

Ambient airborne particulate matter (PM) has diverse sources and physicochemical properties. It affects the transport, transformation and deposition of chemical species, and has significant impacts on radiative forcing and on human health (Pöschl, 2005; USEPA, 2009). The elemental and organic carbon (EC and OC) components constitute a substantial proportion of total particle mass (USEPA, 2009; Putaud et al., 2010; AQEG, 2012). However, the characterisation and source apportionment of the organic component remains a major challenge (Fuzzi et al., 2006; Hallquist et al., 2009; Jimenez et al., 2009). Understanding the sources of this organic aerosol (OA) is important in order to devise effective reduction strategies for ambient PM concentrations (Heal et al., 2012).

Organic aerosol is typically a complex mixture of thousands of organic species, the majority of which are present at low concentrations (less than a few $ng\,m^{-3}$). Current levels of scientific understanding, instrumentation and modelling capability mean that explicit measurement and modelling of all individual OA species is not feasible at present. Measurement of OA by on-line mass spectrometry, such as with the Aerodyne Aerosol Mass Spectrometer (AMS; Canagaratna et al. (2007)), and consideration of individual organic marker ions coupled with multivariate statistical techniques such as positive matrix factorization (PMF; Paatero and Tapper (1994); Paatero (1997)), have facilitated the subdivision of the OA component into empirical categories. These include hydrocarbon-like organic aerosol (HOA), low-volatility and semi-volatile oxygenated organic aerosol (LV-OOA, SV-OOA), solid-fuel organic aerosol (SFOA), cooking organic aerosol (COA) and a number of other categories (Ulbrich et al., 2009; Ng et al., 2010; Lanz et al., 2010; Ng et al., 2011; Young et al., 2015a).

Even allowing for the uncertainties in defining and measuring OA components, there is a general tendency for atmospheric chemistry transport model (ACTM) simulations to underestimate observed amounts of OA and SOA. For example, the AeroCom (Aerosol Comparisons between Observations and Models) project, which includes ~30 global ACTMs and global circulation models (GCMs), has concluded that the amount of OA present in the atmosphere remains largely underestimated (Tsigaridis et al., 2014). Similarly, in an evaluation of 7 global models, Pan et al. (2015) reported a systematic underestimation of OA over South Asia.

Several regional ACTM studies have also reported an underestimation of total OA (Simpson et al., 2007; Murphy and Pandis, 2009; Hodzic et al., 2010; Aksoyoglu et al., 2011; Jathar et al., 2011; Bergström et al., 2012; Koo et al., 2014) and SOA (Hodzic et al., 2010; Shrivastava et al., 2011; Zhang et al., 2013; Fountoukis et al., 2014), with normalised mean biases (NMB) often in the range of $-30\%$ to $-50\%$. In some cases, this underestimation has been shown to be due to problems with the underlying emission inventories, particularly for domestic wood-burning in wintertime (Simpson et al., 2007; Denier van der Gon et al.,





2015). There may also be sources of biogenic secondary organic aerosol (BSOA) arising from previously neglected VOC emissions such as those induced by biotic stress (Berg et al., 2013; Bergström et al., 2014).

Currently, ACTMs cannot explicitly simulate all the kinetic and thermodynamic processes associated with the evolving gas-phase chemistry of semi-volatile organic compounds and their partitioning to the particle phase (Donahue et al., 2014). A widely used heuristic parametrisation for simulating OA is the volatility basis set (Donahue et al., 2011, 2012). The volatility

(in this case the saturation concentration, C*) of gas-phase organic compounds are sorted into bins: low volatility organic compounds (LVOCs, C* $\leq$ 0.1 µg m$^{-3}$; with no lower C*, this category also incorporates extremely-low-volatility organic compounds, ELVOC), semi-volatile organic compounds (SVOCs, C* = 1–10$^3$ µg m$^{-3}$), intermediate volatility organic compounds (IVOCs, C* = 10$^4$–10$^6$ µg m$^{-3}$) and volatile organic compounds (VOCs, C* $\geq$ 10$^7$). Thus, organic compounds are distributed across a continuum from particles to gases. Under typical ambient conditions, all LVOCs, some of the SVOCs, and

essentially none of the IVOCs or VOCs are in the condensed phase (Donahue et al., 2006).

Current emissions inventories, however, only report estimates for VOCs (C* $\geq$ 10$^7$ µg m$^{-3}$) and for the particle fraction of the emissions of species with lower volatilities. The main reason for this is that compounds with intermediate volatility (SVOCs and IVOCs) are difficult to quantify and this is currently not routinely undertaken alongside the techniques that have been developed to measure the more volatile gases (e.g., gas chromatography) or organic-containing particles (e.g., aerosol

mass spectrometry).

Robinson et al. (2007) and Shrivastava et al. (2008) estimated the mass of emitted IVOCs to be 1.5 times that of POA emissions. In their study, this addition of IVOCs = 1.5×POA was applied to all sources of POA – from diesel to biomass burning. They based this estimation on chassis dynamometer tailpipe measurements by Schauer et al. (1999). Since then, several regional and global ACTM applications have adopted this factor of 1.5 (e.g., Murphy and Pandis (2009); Tsimpidi

et al. (2010); Hodzic et al. (2010); Jathar et al. (2011); Fountoukis et al. (2011); Genberg et al. (2011); Zhang et al. (2013); Bergström et al. (2012)). A number of studies, including many of those cited above reporting model underestimation of OA, have highlighted the need for improved measurement and speciation of SVOCs and IVOCs and for these species to be reported in inventories.

Jathar et al. (2014) performed emissions and smog chamber experiments on SOA production from gasoline and diesel vehi-

cles. Diesel contains hydrocarbons with a higher carbon number (C$_8$–C$_{20}$) than gasoline (mainly C$_4$–C$_{10}$). The typical method used for hydrocarbon analysis is gas chromatography (GC); however as the carbon number increases, the number of potential structural isomers increases exponentially, meaning GC is unable to distinguish individual species in the intermediate volatility range (Goldstein and Galbally, 2007). The total carbon of this unresolved complex mixture was estimated and Jathar et al. concluded that these unspeciated organic gases dominate SOA production compared with SOA from the speciated precursors

commonly included in emissions inventories (single-ring aromatics, isoprene, terpenes and large alkenes). Jathar et al. (2014) also performed box-model simulations of the SOA budget for the US, with the addition of unspeciated emissions based on measurements by Schauer et al. (1999), and concluded that gasoline contributes much more to SOA than does diesel. This result is similar to that of Bahreini et al. (2012) who, based on measurements in the Los Angeles Basin, California (CA), concluded that the contribution of diesel emissions to SOA was zero within measurement uncertainty. Conversely, Gentner et al.



(2012) reported that diesel was responsible for 65-90% of vehicular-derived SOA based on measurements of gas-phase organic carbon in the Caldecoff Tunnel, CA, and in Bakersfield, CA, and on estimations of SOA yields. The huge dissimilarity in these conclusions, even in the same state in the US, emphasizes the need for continued research into gasoline- and diesel-related SOA formation. Furthermore, the US and Europe have very different diesel vehicle profiles: in the US, a negligible proportion of passenger cars are diesel (3%), whilst on average across Europe 33% of passenger cars are diesel and this proportion is in-

creasing (Cames and Helmers, 2013). Globally, the demand for diesel fuel is increasing and by 2020 it is expected to overtake gasoline as the principal transport fuel used worldwide (Exxon Mobil, 2014).

In this study, we present new high-resolution simulations of SOA formation in a 3-D ACTM model which includes additional diesel-related IVOC emissions derived directly from comprehensive field measurements of IVOCS and VOCs at an urban background site in central London (Dunmore et al., 2015) during the Clean Air for London (ClearfLo) campaign in 2012

(Bohnenstengel et al., 2014). Modelled concentrations are compared with OA components derived by PMF analysis of AMS measurements during the same campaign, including comparisons with the long-term measurements (full year) as well as the two month-long Intensive Observation Periods (IOPs) in winter and summer.

## 2 Methods

### 2.1 Model description

The EMEP4UK model is a regional application of the EMEP MSC-W (European Monitoring and Evaluation Programme Meteorological Sythesizing Centre-West) model. The EMEP MSC-W model is a 3-D Eulerian model that has been used for both scientific studies and policy making in Europe. A detailed description of the EMEP MSC-W model, including references to evaluation and application studies is available in Simpson et al. (2012), Schulz et al. (2013), and at www.emep.int. The EMEP4UK model is described in Vieno et al. (2010, 2014), and the model used here is based on version v4.5.

The EMEP4UK model uses one-way nesting from a 50 km × 50 km greater European domain to a nested 5 km × 5 km area covering the British Isles and parts of the near continent. The model has 21 vertical levels, extending from the ground to 100 hPa. The lowest vertical layer is ~40 m thick, meaning that modelled surface concentrations represent the average for a 5 km × 5 km × 40 m grid cell. The model time-step varies from 20 s (chemistry) to 5 min and 20 min for advection in the inner and outer domains, respectively.

The model was driven by output from the Weather Research and Forecasting (WRF) model (www.wrf-model.org, version 3.1.1) including data assimilation of 6-hourly model meteorological reanalysis from the US National Center for Environmental Prediction (NCEP)/National Center for Atmospheric Research (NCAR) Global Forecast System (GFS) at 1° resolution (NCEP, 2000).



## 2.2 Emissions

Gridded emissions of $NO_x$ (NO + $NO_2$), $SO_2$, $NH_3$, CO, NMVOCs (Non-Methane VOCs), $PM_{2.5}$ (PM with diameter < 2.5 µm) and $PM_{10}$ (PM with diameter < 10 µm) were obtained from NAEI (National Atmospheric Emissions Inventory, NAEI (2013) for the UK and from CEIP (EMEP Centre on Emission Inventories and Projections; CEIP (2015)) for the rest of the model domain. All emissions are apportioned across a standard set of emission source sectors, following the sector structure defined in the Selected Nomenclature for Air Pollutant (SNAP; EEA (2013); Table 1), consistently applied across the whole domain. In recent years, the Nomenclature for Reporting (NFR) has replaced SNAP categories for official emission reporting

of parties under the Convention on Long-range Transboundary Air Pollution (CLRTAP). This was agreed in an attempt to harmonise reporting requirements between air pollutant and greenhouse gas emission reporting obligations (for instance, the protocols under CLRTAP and the Intergovernmental Panel on Climate Change). The resulting sectoral structure, however, is more aggregated than SNAP and does not allow for equally detailed analyses of individual source types with specific emission characteristics (e.g., fuel types, technologies, temporal emission patterns). Hence, emission datasets for ACTMs are typically

still compiled to reflect SNAP sectors.

Primary PM emissions reported as $PM_{2.5}$ and $PM_{10}$ in the NAEI and CEIP were speciated into EC, OA from fossil fuel combustion, OA from domestic combustion and remaining primary PM by source sectors (using splits developed by Kuenen et al. (2014); as in Fig. 1). Organic matter to organic carbon ratios (OM/OC) of 1.25 and 1.70 are used for HOA and SFOA, respectively (as in Bergström et al. (2012), based on Aiken et al. (2008)). Default speciation of NMVOC emissions into 14

surrogate groups was used (Simpson et al., 2012). International shipping emissions from Entec UK Limited (now Amec Foster Wheeler) were used (Entec, 2010). The annual sectoral total emissions are temporally distributed to hourly resolution using hour-of-day, day-of-week and monthly emission factors for each source sector as incorporated in the EMEP ACTM (Simpson et al., 2012). Daily emissions of all the aforementioned trace gases and particles from natural fires were taken from the Fire INventory from NCAR version 1.0 (FINNv1, Wiedinmyer et al. (2011)). Monthly $NO_x$ emissions from in-flight aircraft, soil

and lightning, as well as biogenic emissions of dimethyl sulphate (DMS), are included as described in Simpson et al. (2012). Biogenic emissions of isoprene and monoterpenes are calculated by the model for every grid cell and time-step. Estimated emissions of wind-blown dust and sea salt are also included but these have no impact on the model simulations of OA (Simpson et al., 2012).

## 2.3 SOA production in the model

The EMEP MSC-W model uses the 1-D volatility basis set (VBS; Donahue et al. (2006)) approach for SOA formation, ageing and phase partitioning. The implementation of the VBS framework within the model, including various options for the treatment of volatility distributions and ageing reactions is described by Bergström et al. (2012).

In the model set-up used here, POA is treated as non-volatile and inert, as is currently assumed by emissions inventories. Having POA be non-volatile allows us to better identify and isolate the SOA formed from our additional diesel IVOCs. Fur-

thermore, it has been demonstrated by Shrivastava et al. (2011) that a 2-species VBS simulates an evolution of oxygen:carbon



ratios (O:C) similar to the 9-species VBS approach. Shrivastava et al.'s two bins were of volatility 0.01 and $10^5$ which, because material with the lower volatility is always completely in the particle phase under ambient conditions, is similar to our non-volatile treatment of POA.

Five volatility bins (C* = 0.1, 1, 10, 100, 1000 $\mu g\,m^{-3}$) are used for SOA from anthropogenic and biogenic VOCs. The SOA yields for alkanes, alkenes, aromatics, isoprene and terpenes under high and low $NO_x$ conditions were taken from Tsimpidi et al. (2010). SOA from alkanes, alkenes and terpenes is assumed to have an initial OM/OC ratio of 1.7; SOA from isoprene 2.0; and SOA from aromatics 2.1 (Bergström et al., 2012; Chhabra et al., 2010). Both anthropogenic SOA (ASOA; from alkanes, alkenes and aromatics) and BSOA (from isoprene and terpenes) undergo atmospheric ageing by the hydroxyl (OH) radical in the model (with rate coefficient of $4.0 \times 10^{-12}\,cm^3\,molecule^{-1}\,s^{-1}$; Lane et al. (2008)), resulting in a shift into the next lower volatility bin and a mass increase of 7.5%.

A constant background OA of $0.4\,\mu g\,m^{-3}$ is used to represent the contribution of OA sources not explicitly included in the model (e.g., oceanic sources or spores). This background OA is assumed to be highly oxygenated (with an OM/OC ratio of 2.0) and is therefore included under modelled SOA.

## 2.4 Additional IVOCs from diesel

Current emissions inventories report highly-volatile anthropogenic VOCs of C* $\geq$ $10^7$ $\mu g\,m^{-3}$ (Passant, 2002). However, diesel vehicles also produce substantial emissions of species with intermediate volatility in the range $10^5 \leq C* \leq 10^6$ $\mu g\,m^{-3}$ (IVOCs), as has been shown by Dunmore et al. (2015) from measurements made at an urban background site in central London during the ClearfLo project.

In this study, IVOC emissions from diesel vehicles were introduced into the model proportionally to on-road transport VOC emissions, using $n$-pentadecane ($C_{15}H_{32}$) as surrogate for the following reasons. First, the amount of alkenes in diesel fuel is low (< 5 %; Gentner et al. (2012)), so an alkane is the most appropriate surrogate. Second, all $n$-alkanes up to $n$-dodecane were individually speciated and quantified during two month-long Intensive Observation Periods (IOPs) during the ClearfLo project and there were strong correlations between all $n$-alkanes that have a predominately diesel source (Dunmore et al., 2015). Third, the rate constant for the linear alkane is a reasonable representation of the rate constant for all the (un-measured) branched and cyclic isomers, as demonstrated by Dunmore et al. (2015) for the $C_{12}$ $n$-alkane, dodecane. The bulk of diesel emissions, however, are likely to have higher carbon numbers than were measured by the GC×GC system. The rate coefficient for the reaction between $n$-pentadecane and OH has been measured in a number of studies ($k = 2.07 \times 10^{-11}$ $cm^3\,molecule^{-1}\,s^{-1}$) unlike for the majority of branched isomers in this range. Furthermore, measurements of diesel fuel composition have shown that the average carbon number on a percentage weight basis was 14.94 (Gentner et al., 2012), so $n$-pentadecane was considered to be an appropriate surrogate for diesel emissions in general.

In the NAEI, emissions from gasoline vehicles dominate the NMVOCs emissions from road traffic, but measurements during the ClearfLo winter Intensive Observation Period showed that NMVOCs assigned to diesel vehicles dominated traffic-related NMVOC concentrations. The amount of pentadecane emitted in the model was therefore set to match the measured diesel-(I)VOCs (IVOCs + VOCs) to gasoline-VOCs ratio (Table 2, Fig. 2). This pentadecane addition was then applied to every





country in the model domain using the same factor as for the UK. This first approximation is justified because the fleet share of diesel vehicles in the UK is similar to the European average (~30%; EEA (2010)), but it can introduce errors for specific countries.

For the oxidation products of $C_{15}H_{32}$, SOA mass yields were taken from Presto et al. (2010): 0.044, 0.071, 0.41, 0.30 for the 0.1, 1, 10, 100 $\mu g\,m^{-3}$ bins, respectively. These yields are reported for SOA with unit density (1 $g\,cm^{-3}$). In this work, SOA density was assumed to be 1.5 $g\,cm^{-3}$ (Tsimpidi et al., 2010; Bergström et al., 2012) and the yields were increased accordingly.

For the UK, our approach adds 90 Gg of diesel-IVOCs emission for the year 2012 (Fig. 2). The 1.5xPOA approach (Shri-
vastava et al. (2008) based on measurements by Schauer et al. (1999)) would only add 31 Gg (Fig. 1). Part of this discrepancy could be attributable to the different methods and circumstances used to derive the additions (this work: five weeks of ambient measurements in a megacity; previous estimate: tailpipe laboratory measurements using different instruments). Another possible reason for the difference is an underestimate in POA emissions in the inventory; more POA would increase the amount of proportionally added IVOCs. However, Dunmore et al. (2015) show that lower carbon number (and higher volatility) NMVOCs
measured during the ClearfLo campaign were consistent with emissions estimates. This lends confidence to adding IVOCs proportionally to reported NMVOC emissions, rather than proportionally to POA emissions. Nevertheless, we have also performed a model run using the POA-based IVOC estimate. The emitted 1.5xPOA IVOCs are assigned to the VBS bin of $10^5$, where they start ageing by reaction with OH ($k = 4.0 \times 10^{-11}$ $cm^3\,molecule^{-1}\,s^{-1}$, Shrivastava et al. (2008)). In this case, the additional IVOCs were calculated from POA from all sources, not just traffic-related. Note that in the UK, most of the additional IVOCs
of the POA-based approach would come from SNAP2 (Residential and non-industrial combustion emissions; (Fig. 1)): 18 Gg, whereas only 5 Gg would be added to SNAP7 (Road transport; and 8 Gg to remaining sectors). Due to the very different absolute amounts and source categories (the latter of which also leads to differences in the spatial pattern and temporal variation of the additional emissions), detailed comparison of the two different additions is not justified, and only annual total ASOA budgets of the different addition methodologies are presented.

## 2.5  Summary of model experiments

Three runs of the EMEP4UK modelling system were performed for 2012:

- Base: all anthropogenic emissions as in officially reported inventories; biogenic emissions calculated by the model for each advection time step.

- addDiesel: Base + additional diesel IVOCs added proportionally to NMVOC emissions from traffic (2.3xSNAP7). The
additional IVOCs were treated using *n*-pentadecane as surrogate species. The semi-volatile VBS-species formed after oxidation of *n*-pentadecane were treated in the same way as the ASOA-species from VOC-oxidation (the same ageing rate and mass increase due to oxygen addition; see Sect. 2.3).

- add1.5xPOA: Base + additional IVOCs added proportionally to all POA emissions (1.5xPOA; as in Shrivastava et al. (2008) based on measurements by Schauer et al. (1999)).



## 2.6 Comparison with measurements

Modelled $OA_{2.5}$ (OA with diameter $< 2.5\ \mu m$) is compared with non-refractory submicron (NR-PM$_1$) OA measured by Aerodyne AMS instruments at an urban background site in central London and at two rural sites shown in Fig. 3 (Xu et al., 2015b; Young et al., 2015a, b; Bohnenstengel et al., 2014). The error introduced to the comparison by the different size fractions is believed to be small, as measurements at an urban background site in Birmingham, England have shown that 90% of organic carbon in $PM_{2.5}$ is in the submicron fraction (Harrison and Yin, 2008).

Different types of AMS were deployed in this campaign. At the London North Kensington site a compact time-of-flight AMS (cToF-AMS) was deployed for a full calendar year (January 2012 – January 2013), and a high-resolution time-of-flight AMS (HR-ToF-AMS) was also deployed for the IOPs at the same site. A HR-ToF-AMS was deployed in Detling during the winter IOP, and in Harwell during the summer IOP. PMF analysis was applied to each of the datasets to apportion measured OA into different components (Ulbrich et al., 2009). A detailed description of the derivation and optimization of the factors retrieved from the AMS data at Detling can be found in Xu et al. (2015b), at London North Kensington in Young et al. (2015a) and Young et al. (2015b), and at Harwell in Di Marco et al. (2015). The OM/OC ratios for each of the PMF datasets presented in this study were calculated with the Improved-Ambient method from Canagaratna et al. (2015). A summary of the instruments, measurement periods and resolved PMF factors is given in Table 3. As European emissions inventories do not currently include cooking OA (COA), this factor could not be compared with the model.

When AMS measurements and their PMF apportionments are compared, some disagreement is observed, even for the two instruments measuring at the same time at the same location at London North Kensington. This is in part due to the differences in the types of AMS used, where more chemical information is retrieved from the HR-ToF-AMS, which can subsequently lead to differences in the derived PMF factors from the individual datasets. It should also be kept in mind that PMF was run on each of the full datasets, covering a full year for the cToF-AMS and only four weeks for each of the HR-ToF-AMS IOPs, thus it is not necessarily expected that the same PMF factors would be derived from the different datasets. Nevertheless, strong correlations between daily averaged primary OA components from the two instruments are observed (0.95, 0.92, and 0.88 for HOA, SFOA, and COA, respectively), with less strong correlations for SOA (0.77). Scatterplots of PMF derived OA component concentrations resolved for the cToF-AMS data and HR-ToF-AMS (winter IOP) are shown in Fig. 4. This inherent uncertainty in the measurements constrains the expected correlation with the model.

The following numerical metrics were used for model evaluation: FAC2 (Factor of 2) - the proportion of modelled concentrations that are within a factor of 2 of the measured concentrations; NMB - normalised mean bias; NMGE - normalised mean gross error; r - correlation coefficient (Carslaw and Ropkins, 2012); and COE - coefficient of determination (Legates and McCabe, 2013), which is defined as:

$$COE = 1.0 - \frac{\sum_{i=1}^{n} |M_i - O_i|}{\sum_{i=1}^{n} |O_i - \overline{O}|}, \tag{1}$$





where $M_i$ is the modelled value, $O_i$ is the corresponding measured value and $\overline{O}$ is the mean measured value. A COE of 1
indicates perfect agreement between model and measurements. Although the COE does not have a lower bound, a negative
COE value indicates that the model is less effective at capturing the variation in measurements than the measurement mean.

NCAR command language (NCL) was used to produce the maps (NCAR, 2015), and R, openair and ggplot2 for the analysis
and all other plots (R Core Team, 2014; Carslaw and Ropkins, 2012; Wickham, 2009). Seasons are defined as follows: winter
- Dec-Jan-Feb (DJF); spring - Mar-Apr-May (MAM); summer - Jun-Jul-Aug (JJA), and autumn - Sep-Oct-Nov (SON).

The configuration of the underlying meteorological model (WRF) used for this study is identical to that described in Vieno
et al. (2010) where it is shown to perform very well in comparison with measurements. No further evaluation is presented here.

## 3   Results

The comparisons between the model results and measurements are presented in the following order. First, comparisons are
presented for primary OA, $NO_x$, $O_3$, and for secondary inorganic aerosol (SIA) to give an overview of the overall performance
of the modelling system. Second, the hourly concentrations of SOA during the two IOPs are evaluated, demonstrating the
agreement between the model and measurements at high temporal resolution. Third the year-long daily SOA concentrations
are compared and the relative impact of diesel-VOCs on SOA production in London is shown. Fourth, modelled and measured
OM/OC ratios are shown, and finally, annual total ASOA from our method and the previous 1.5xPOA approach are compared.

### 3.1   POA, $NO_x$, $O_3$, SIA: annual dataset

Figure 5 shows the year-long comparison between the daily-averaged model results and the cToF-AMS measurements at the
London North Kensington site. The model underestimates primary OA (HOA and SFOA) concentrations (NMB of $-54\%$
and $-71\%$, respectively), but shows good daily correlations (r-values of 0.53 and 0.72, respectively). The underestimation
of HOA may be caused by a combination of lack of model resolution (e.g., the minor road close to the measurement site
can not be resolved with the 5 km grid), and underestimation of PM emissions. Modelled $NO_x$ concentrations are relatively
less underestimated in comparison to measurements (NMB of $-32\%$, Fig. 6a), suggesting that HOA emissions may be more
underestimated than the emissions of $NO_x$. Concentrations of secondary pollutants are simulated well by the model in the
gas-phase (Fig. 6b, with a NMB of $-1\%$ for ozone), and for inorganic PM constituents (Fig. 6c–d), with NMBs of $6\%$ for
$SO_4^{2-}$, $-12\%$ for $NH_4^+$, and $-23\%$ for $NO_3^-$.

### 3.2   Hourly comparison of secondary OA: summer IOP

Evaluation statistics between hourly measured and modelled SOA concentrations in July and August 2012 (summer IOP)
show excellent agreement (Fig. 7). The values of $r$ for the Base run were 0.67 and 0.55 at North Kensington and Harwell
respectively. The addDiesel experiment yields a modest improvement in the value of $r$ at North Kensington (to 0.76) and a
marked improvement in Harwell (to 0.74). The addDiesel run substantially improves the NMB for SOA at the Harwell and
London North Kensington sites from $-32\%$ to $-5\%$, and from $-35\%$ to $0.1\%$, respectively (Fig. 7). This means that ~30%





of SOA at both sites during this period can be explained by the diesel IVOCs added into the model using pentadecane as a
surrogate. There is also marked improvement of model-measurement COE values at the two sites (Harwell, 0.26 to 0.42, and
NK 0.31 to 0.45). The improvement in NMGE is noticeable (Harwell, 54% to 43%, and NK 59% to 47%), but smaller than the
improvements in the other metrics. It can be seen from the scatter-plots in Fig. 8 that most modelled hourly SOA concentrations
fall within a factor of two of the measured concentrations (FAC2 for the addDiesel experiment is 78% at Harwell and 62% at
NK).

Measured and modelled mean hour of day variations of SOA concentrations are presented in Fig. 9, where it can be seen
that measured SOA concentrations do not have a very strong diurnal cycle. Interestingly, both sites exhibit dips in measured
SOA concentrations in the morning and early evening. Both measured and modelled SOA concentrations in London North
Kensington reach a maximum in the afternoon, but SOA of the addDiesel experiment starts this increase earlier than the
measurements, meaning that our ASOA production from pentadecane might be too rapid.

During the summer IOP, there were two sustained episodes of increased SOA concentrations: 23-Jul to 28-Jul and 9-Aug to
13-Aug. Only London North Kensington had measurements during the first episode and the elevated concentrations were well
captured by the addDiesel simulation (including the highest peak of greater than 16 $\mu$g m$^{-3}$: 27-Jul 13:00, Fig. 10b). Daily
averaged SOA maps (Fig. 11) suggest that this first episode arose from a combination of SOA transported from Europe and
SOA produced locally in London. A region of elevated concentration around London exists within a general gradient of SOA
from continental Europe to Southern England. Even daily averaged concentrations are spatially variable during this episode
meaning that inaccuracies in some of the modelled peaks can be attributed to uncertainties in the underlying meteorological
model. Most of the modelled SOA during this episode was of anthropogenic origin with the addDiesel run yielding a significant
portion of ASOA from pentadecane.

For the second sustained episode of high SOA concentrations, from 9-Aug to 13-Aug, several features remain substantially
underestimated even in the addDiesel run. For Harwell, the model does capture two of the highest peaks (10-Aug 22:00 mea-
sured: 6.8 $\mu$g m$^{-3}$, addDiesel: 8.5 $\mu$g m$^{-3}$ and 12-Aug 12:00 measured: 7.9 $\mu$g m$^{-3}$, addDiesel: 7.0 $\mu$g m$^{-3}$), but for London
North Kensington, the model simulates a minimum during the highest measured concentration (10-Aug 05:00 measured: 11.9
$\mu$g m$^{-3}$, addDiesel: 2.0 $\mu$g m$^{-3}$). The high concentrations during the first two days of this episode were very localised with
horizontal widths of just tens of kilometres (Fig. 12a,b). There was a build-up of pollution caused by high pressure and low
boundary-layer height (BLH), which led to production of ASOA in London. The high variability in the modelled concentra-
tions (for example, the simulated minimum during the measured maximum at North Kensington) is caused by the shifting of
this narrow ASOA plume in space (Fig. 10b). On 12-Aug, this episode was also subject to SOA contribution from Europe
(Fig. 12d).

During the period of overlapping measurements at Harwell and North Kensington (3-Aug–18-Aug), both the measurements
and the model agree with a modest rural to urban increase. Average measured SOA concentrations were 2.4 $\mu$g m$^{-3}$ and 2.6
$\mu$g m$^{-3}$ for Harwell and North Kensington, respectively, whilst average modelled concentrations were 2.3 $\mu$g m$^{-3}$ and 2.5
$\mu$g m$^{-3}$ (for the addDiesel experiment).



### 3.3 Hourly comparison of secondary OA: winter IOP

Both the Detling and London North Kensington sites exhibit good modelled-measured hourly correlation ($r$ = 0.63 and 0.64, addDiesel run; Fig. 13). The addDiesel run decreases the NMB for SOA at these sites from $-59\%$ to $-30\%$ for Detling, and from $-24\%$ to $8\%$ for London North Kensington. This means that ~30% of SOA at these sites during this period can be explained by diesel IVOCs. In Detling, there is also a pronounced improvement in the COE, from 0.10 to 0.31. In North Kensington, the COE was already high but is increased from 0.27 to 0.30. It can be seen in Fig. 13 as well as Fig. 14 that lower concentrations of SOA (19-Jan–27-Jan) are overestimated by the model. This overestimation is caused by the very simplified method of including missing sources of OA using a constant concentration of $0.4\ \mathrm{\mu g\,m^{-3}}$ (which is assumed to be highly oxygenated and is therefore included under modelled SOA). As a constant, this background OA does not currently go through atmospheric emission-removal processes in the model. However, the period in question exhibited snowfall, removing much

of the aerosol (as can be seen from the very low concentrations measured in both Detling and London North Kensington). Nevertheless, this does not mean that the background OA concentration is an overestimate - this simplified inclusion is set for the whole European domain and regardless of season. Explicit inclusion of additional missing biogenic sources of OA to the model is already part of ongoing development of the model and will be presented in future studies.

   During the ClearfLo Winter IOP, measured SOA concentrations were higher in Detling than in North Kensington (Fig. 13).

This is correctly captured by the simulations and is caused by a steep positive gradient of concentrations from southern England across to the near European continent (Fig. 15). The measured Detling/North Kensington SOA ratio (ratio of average concentrations for this period) was 1.8 while the modelled ratio was 1.1, so the model correctly simulates the direction of the spatial gradient, but underestimates its magnitude. For North Kensington, the model also captures that SOA concentrations are lower on Feb-5 than on Feb-4. In Detling, however, measured concentrations were higher on Feb 5, which the model does not

reproduce. During the night of 4-5 February, the wind was very strong ($> 10\ \mathrm{m\,s^{-1}}$) and there was a small shift between the measured wind direction and the wind direction input to EMEP4UK from WRF. As a consequence, the simulated pollution plume was shifted too much to the east (Fig. 15b) causing the model-measurement discrepancy on this particular occasion.

   Even though the additional diesel IVOCs noticeably increased the modelled SOA concentrations during the winter IOP, there is still a marked underestimation of elevated measured SOA concentrations during 15-Jan–19-Jan and 30-Jan–4-Feb. During

these periods, the observed temperature was colder than the average temperature of the winter IOP (Crilley et al., 2015) and peaks in measured SOA also coincide with elevated concentrations of SFOA (Figs. 5b and 13b). As our modelled SFOA is underestimated by a factor of 4 (NMB of $-72\%$), it is likely that (i) SOA precursor VOC emissions from domestic heating are also underestimated, and (ii) adding missing IVOCs from this emission sector would contribute to the modelled SOA during these periods. It has been recently shown by Denier van der Gon et al. (2015) that the emission factors used by different

European countries for wood combustion PM emissions, even for the same appliance type, can differ by a factor of 5. They constructed a revised inventory, in which each country's emission was updated using an unified emission factor. This resulted in increases of PM (and estimated accompanying IVOC) emission estimates for most countries.





### 3.4 Daily and seasonal secondary OA: annual dataset

Time-series of daily averaged modelled and measured SOA concentrations for the whole year are shown in Fig. 16. Table 4
gives daily modelled vs measured SOA evaluation statistics during different seasons at the North Kensington site. Values for
autumn are presented with and without the two extreme points (size of the data set $n = 91$ and $n = 89$).

For the daily model-measurements comparison, spring has the highest correlation ($r = 0.85$, both Base and addDiesel;
Table 4). This can also be seen from the time series (Fig. 16: March–May) where both model simulations follow most of
the measured peaks. The Base run $r$-value for spring was already high, but nevertheless, the addDiesel run shows a marked
improvement for all other model evaluation statistics. FAC2 is increased by 10%, COE is increased to 0.39, NMB is reduced
by 35% and NMGE is reduced by 7%. The NMGE of 38% remaining in the addDiesel model run is probably governed by
uncertainties in meteorology, as well as by uncertainties in the temporal and spatial variability of emissions. During summer, the
model captures the majority of the periods of increased SOA mass well (e.g., Jun-28, Jul-22 - Jul-29, Aug-15, Aug-20, Fig. 16:
June–August), but there is some model underestimation when SOA concentrations were lower ($< 2\,\mu g\,m^{-3}$). As for spring,
the addDiesel experiment improves all model evaluation statistics. More detailed hourly analysis of the SOA concentrations
during the summer IOP (end of July to August) was presented in Sect. 3.2.

The model performance is less good in autumn than during the other seasons. There are some days where the Base case
scenario overestimates measured SOA (23–25-Oct, 21-Nov, 24-Nov) with the addDiesel run increasing this further. During
these days, particle nitrate ($NO_3^-$) and ammonium ($NH_4^+$) are also substantially overestimated by the model (Fig. 6). This
suggests that the overestimations are likely caused by errors occurring during this period in the meteorological forecasts, e.g.,
missed rain events, rather than by uncertainties in the formation of secondary organic aerosol specifically.

The model evaluation statistics for autumn are strongly influenced by the two modelled values on 23-Oct and 24-Oct (Ta-
ble 4). Removing these two values reduces the seasonal average SOA concentration modelled with the addDiesel run by 33%
($2.0$ and $1.5\,\mu g\,m^{-3}$ with and without these two points, respectively). Their combined influence on the annual average modelled
concentration is 8%, which is substantially more than any other points of the annual dataset.

For the winter months, modelled concentrations in January are much lower than measurements, whereas in February the
timing of several peaks is well reproduced and even overestimated by the addDiesel experiment. Detailed hourly analysis of
the SOA concentrations during the winter IOP has been presented in Sect. 3.3. In December, measured SOA concentrations
were much lower than in January and even though the model captures the highest peak, there is some overestimation in the
lowest range ($< 0.5\,\mu g\,m^{-3}$).

Figure 17 shows annually and seasonally averaged measured and modelled SOA. The difference between the Base and
addDiesel experiments illustrate the impact of missing IVOC emissions from diesel-traffic on SOA formation. As was discussed
before, and can be seen from Table 4, IVOC precursors from diesel vehicles reduce the NMB by ~30%, which as an annual
average is $0.6\,\mu g\,m^{-3}$ of additional SOA. Moreover, the 90-th percentile of daily averaged SOA concentrations of the addDiesel
experiment is $3.8\,\mu g\,m^{-3}$ (which is similar to the measured 90th percentile of $3.2\,\mu g\,m^{-3}$), whereas the 90-th percentile of
the Base case simulation is $2.2\,\mu g\,m^{-3}$. This means that (i) on 36 days of the year, SOA is a notable component of PM (the



annual average $PM_{2.5}$ concentration limit value of the European Union Directive 2008/50/EC is 25 µg m$^{-3}$), and (ii) during
those days, the relative contribution to SOA from diesel IVOCs could be greater than 40%.

### 3.5 OM/OC ratios

Measured OM/OC ratios for SOA were generally higher than those modelled (1.99–2.34 vs 1.88–1.97, Table 5). Nevertheless,
the measured OM/OC ratio at London North Kensington during the summer IOP was the lowest of the measured range: 1.99,
which is a close match to modelled SOA OM/OC ratio for that period: 1.97. Model performance for spring and summer was
shown to be very good, but it is possible that the missing SOA precursors in the colder months (from domestic heating) could
yield SOA with higher initial OM/OC ratios, thereby increasing the annual average value. Furthermore, wintertime simulations
of SOA in Paris by Fountoukis et al. (2015) also showed large underestimations and they speculated that this could be pointing
towards an SOA formation process during low photochemical activity periods that is currently not simulated in atmospheric
chemistry transport models.

### 3.6 Comparison to the previous (IVOCs=1.5xPOA) approach

Figure 18 shows the annual average ASOA concentrations at London North Kensington modelled with different assumptions
for additional IVOC emissions. As was explained in Sect. 2.4, for the UK, the addDiesel experiment adds 90 Gg of diesel-
related IVOCs proportionally to road transport emissions (SNAP7), whereas the IVOCs=1.5xPOA approach only adds 5 Gg
to SNAP7 and another 26 Gg to other sectors (mainly to SNAP2: residential and non-industrial combustion). Therefore, our
approach creates a considerably larger amount of SOA from IVOCs (and only from diesel-related IVOCs) than the previous
method.

## 4 Discussion

We show that 30% of SOA in London could be produced from diesel-related IVOC emissions that are not currently included
in the emissions inventories. To our knowledge, this is the first study where IVOC emissions are added proportionally to
NMVOC emissions (as opposed to addition proportionally to POA emissions). Moreover, previous studies have added IVOCs
proportionally to POA from all sources, whereas this study focuses specifically on the impact of diesel-IVOCs from on-road
traffic emissions (IVOCs = 2.3xSNAP7 VOCs). There is reason to believe that higher volatility VOCs are better represented in
current emissions inventories than the emissions of PM. Also, the official inventories do not provide the individual contribution
of POA to total PM. Therefore, the addition of IVOCs proportionally to NMVOCs may be better constrained than the POA-
based approach used in studies so far. The additional emissions are also tied directly to the relevant emission source category.
   There are several possible uncertainties in our estimate of additional IVOCs, and subsequent SOA production and ageing.
As a first approximation, we added IVOCs to each European country based on our measurements in London. This was justified
as the diesel usage in the UK is similar to the European average. Furthermore, different European countries might be using
different emissions factors for their estimates of NMVOCs from gasoline and diesel or have a different average fleet age than




the UK. Therefore the refinement of this addition should be evaluated in each country's emissions inventory and reported to

CEIP. It should be noted that two of the most populous countries in Europe - France and Germany - both have a higher diesel

penetration than the UK and therefore for western central Europe our addition is rather conservative. We believe it would be

beneficial to further refine the estimate of diesel-IVOCs treating each country separately.

It was seen from the hourly profiles at the London North Kensington site during the summer IOP (Fig. 9b) that both the

model and the measurements exhibit a small diurnal cycle (peaking in the afternoon). Even though somewhat counter-intuitive

(as most of the SOA chemistry is photochemically driven through reaction with the OH radical), an absence of a strong diurnal

cycle of SOA has been seen in many European studies (Zhang et al., 2013; Fountoukis et al., 2014; Young et al., 2015a). A

relatively small daytime increase of SOA could be explained by the expansion of the boundary layer height (Xu et al., 2015a).

PMF measurements of SOA in Mexico City, on the other hand, revealed a very strong diurnal cycle, peaking around the mid-day

(Shrivastava et al., 2011). The fact that during the summer IOP our addDiesel experiment exhibits a slightly stronger diurnal

cycle than the measurements (with day-time values slightly overestimated and night-time underestimated) indicates that the

SOA yields could be too high. We assumed an SOA density of $1.5\,\mathrm{g\,cm^{-3}}$ and increased the yields linearly, as has been done

in all other ACTM studies. Actually, increasing the assumed density of SOA from the unit value ($1\,\mathrm{g\,cm^{-3}}$) changes the total

$C_{OA}$ on the Odum mass yield plots (Odum et al., 1996) used to derive the yields from the chamber experiment. Therefore,

increasing the yields linearly is not exactly correct (Donahue 2015, personal contact) and further studies and refinement into

the calculation of SOA yields and density would be beneficial.

We use an ageing rate of $4.0 \times 10^{-12}\,\mathrm{cm^3\,molecule^{-1}\,s^{-1}}$ for both ASOA and BSOA. This is slower than has been used in

some other studies (for example, Tsimpidi et al. (2010) uses $4.0 \times 10^{-11}\,\mathrm{cm^3\,molecule^{-1}\,s^{-1}}$: 10 times faster, or Fountoukis

et al. (2011) uses $1.0 \times 10^{-11}\,\mathrm{cm^3\,molecule^{-1}\,s^{-1}}$: 2.5 times faster). A combination of lower initial SOA yields, but slightly

higher ageing rates could possibly flatten the diurnal cycle of our modelled SOA, matching the measurements better. Therefore,

an improvement for the detailed, hourly, evolution could be achieved by a sensitivity study of these yields and ageing rates.

This does not, however, change the main scope and results of this paper which illustrate the relative impact of the diesel-IVOCs

on SOA formation.

In the current set-up of the EMEP model, only two PM size fractions are simulated: $PM_{2.5}$ and $PM_{2.5-10}$, because only

two fractions are included in the emissions inventories for PM used in this study. Even though on an annual basis, 90% of

$OC_{2.5}$ is in the sub-micron ($OC_1$) range (Sect. 2.6), the comparison between a modelled $OC_{2.5}$ and a measured $OC_1$ could be

introducing larger errors during specific days or hours. Therefore, as AMS measurements become more prevalent, emissions

inventories should be reported for all three size classes, $PM_1$, $PM_{1-2.5}$, $PM_{2.5-10}$. This would allow the model to partition

SOA into the corresponding fractions, making the direct comparison of modelled $SOA_1$ to measured $SOA_1$ possible.

In the evaluation of modelled and measured SOA, it was shown that some of the uncertainties in the modelled concentrations

are caused by errors in modelled wind vectors. Nevertheless, the underlying meteorological model works well (as demonstrated

by comparisons of different pollutants for the whole calendar year), and overall the errors caused by meteorology are believed

to be relatively smaller than those introduced by emissions (amount, volatility, composition), or SOA yields and ageing rates.



## 5   Conclusions

This study presents annual time series of new high-resolution simulations of SOA formation with the EMEP4UK ACTM (which is a nested application of the EMEP MSC-W model over the British Isles). Our simulations include additional diesel-related IVOC emissions derived directly from comprehensive field measurements of IVOCS and VOCs at an urban background site in central London. To our knowledge, this is the first study where IVOC emissions are added proportionally to VOC emissions (as opposed to proportionally to POA emissions). Moreover, previous studies have added IVOCs proportionally to POA from all sources, whereas this study focusses specifically on the impact of diesel-IVOCs from on road traffic emissions (IVOCs = 2.3xSNAP7 VOCs). This addition of IVOCs proportionally to NMVOCs may be better constrained than the POA-based

5   approach used in studies so far.

Modelled concentrations were compared with OA components derived from PMF analysis of AMS measurements, and four groups of SOA evaluation was presented: (i) hourly comparison during a summer IOP (Intensive Observation Period), (ii) hourly comparison during a winter IOP, (iii) daily comparison for a full calendar year (including seasonal statistics), and (iv) comparison of OM/OC ratios of different OA components. Overall, very good performance in comparison to the measurements was shown, giving us confidence in the SOA prediction skill of the ACTM system used. To our knowledge, this is the first study

where modelled OA components are compared with a year-long dataset of PMF apportioned AMS measurements.

During the period of concurrent measurements, SOA concentrations at the Detling rural background location were greater than at the central London location. The model showed that this was caused by an intense pollution plume with a strong gradient of imported SOA passing over the rural location and demonstrates how short periods of measurements can give a different picture compared with longer-term measurements, as well as the value of atmospheric chemistry-transport modelling

for supporting the interpretation of measurements taken at different sites or for short durations.

It was concluded that diesel IVOCs alone can explain on average ~30% of the annual SOA in and around London. Moreover, the 90-th percentile of modelled daily SOA concentrations for the whole year is 3.8 $\mu g\,m^{-3}$, more than 40% of which is produced from the missing diesel precursors. Therefore, further refinement of these precursors (currently not included in official emissions inventories) is recommended.

*Acknowledgements.* The authors acknowledge the UK Department for Environment, Food and Rural Affairs (Defra) and the Devolved Administrations, through the projects: development of the EMEP4UK model (AQ0727), as well as access to the AURN data, which were obtained from uk-air.defra.gov.uk and are subject to Crown 2014 copyright, Defra, licenced under the Open Government Licence (OGL), and for partial support for the aerosol measurements. Partial support for the EMEP4UK modelling from the European Commission FP7 ECLAIRE project is gratefully acknowledged. This work was supported in part by the UK Natural Environment Research Council (NERC)

ClearfLo project [grant ref. NE/H008136/1] and co-ordinated by the National Centre for Atmospheric Science (NCAS). R. Ots was supported by a PhD studentship (University of Edinburgh and NERC-CEH contract 587/NEC03805). D. E. Young was supported by a NERC PhD studentship [ref. NE/I528142/1]. R. E. Dunmore was supported by a NERC PhD studentship [ref. NE/J500197/1]. NLN, LX, LRW and SCH were supported by the US Department of Energy (grant no.DE-SC000602). The authors would like to thank David Simpson for helpful advice about the EMEP model.



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

**Table 1.** SNAP source sectors as specified in the emissions input to the model (CEIP, 2015).

| | |
|---|---|
| SNAP1 | Combustion in energy and transformation industries |
| SNAP2 | Residential and non-industrial combustion |
| SNAP3 | Combustion in manufacturing industry |
| SNAP4 | Production processes |
| SNAP5 | Extraction and distribution of fossil fuels |
| SNAP6 | Solvent and other product use |
| SNAP7 | Road transport |
| SNAP8 | Other mobile sources and machinery |
| SNAP9 | Waste treatment and disposal |
| SNAP10 | Agriculture |

**Table 2.** Comparison of diesel and gasoline NMVOCs in the UK National Atmospheric Emissions Inventory (NAEI) with the urban background ambient concentrations measured during the ClearfLo winter Intensive Observation Period in London.

| | NAEI 2012 (emission) | Measurements [a] (concentration) |
|---|---|---|
| Diesel-(I)VOCs | $8 \, \mathrm{Gg \, yr^{-1}}$ | $107 \, \mathrm{\mu g \, m^{-3}}$ |
| Gasoline-VOCs | $31 \, \mathrm{Gg \, yr^{-1}}$ | $33 \, \mathrm{\mu g \, m^{-3}}$ |
| Diesel/Gasoline | 0.26 | 3.2 |

[a] Dunmore et al. (2015).



**Table 3.** AMS measurements and resolved PMF factors during the ClearfLo campaign and the allocation of the PMF factors to SOA for comparison with model simulations. Site locations are shown in Fig. 3. Site names are abbreviated as follows: NK - London North Kensington, DET - Detling, HAR - Harwell.

|  |  |  |  | PMF factors | |
| --- | --- | --- | --- | --- | --- |
| Period | Site | Dates (year 2012) | Instrument | Primary | Secondary |
| winter IOP | NK | 13-Jan–8-Feb | HR-ToF-AMS | HOA, SFOA1, SFOA2, COA | OOA |
|  | DET | 20-Jan–14-Feb | HR-ToF-AMS | HOA, SFOA | OOA |
| summer IOP | NK | 21-Jul–19-Aug | HR-ToF-AMS | HOA, COA, Unknown | SV-OOA, LV-OOA, N-OOA |
|  | HAR | 3-Aug–20-Aug | HR-ToF-AMS | HOA | OOA |
| annual | NK | 11-Jan–24-Jan (2013)* | cToF-AMS | HOA, SFOA**, COA | OOA1, OO2** |

* As the cToF-AMS was retuned before the summer IOP and retuned to the previous tuning at the end of the IOP, the subsequent data could not be used in the PMF analysis (see Young et al. (2015a) for details). However, for the purpose of the comparison in this study, data from the HR-ToF-AMS, deployed at the same site during the summer IOP, was used to fill in this period.

** PMF analysis revealed the SFOA and OOA2 factors were convolved due to their similar, strong diurnal cycles. Daily averages have been used to estimate their concentrations (Young et al., 2015a).





**Table 4.** Model-measurements comparison statistics for daily SOA at London North Kensington. Autumn is presented with and without the two outliers (23-Oct and 24-Oct. $n = 91$ and 89, respectively).

| | Base | addDiesel | Base | addDiesel |
|---|---|---|---|---|
| | spring (MAM) | | summer (JJA) | |
| $n$ (days) | 91 | | 86 | |
| FAC2 | 64% | 74% | 60% | 79% |
| NMB | -35% | 0.1% | -34% | -5% |
| NMGE | 45% | 38% | 48% | 39% |
| $r$ | 0.85 | 0.85 | 0.71 | 0.82 |
| COE | 0.29 | 0.39 | 0.26 | 0.41 |
| | autumn (SON) | | winter (JFD) | |
| $n$ (days) | 89 | | 81 | |
| FAC2 | 82% | 74% | 70% | 69% |
| NMB | -2% | 58% | -28% | 6% |
| NMGE | 52% | 96% | 47% | 61% |
| $r$ | 0.38 | 0.28 | 0.40 | 0.40 |
| COE | -0.13 | -1.07 | 0.21 | -0.02 |
| | autumn (SON) | | | |
| $n$ (days) | 91 | | | |
| FAC2 | 80% | 73% | | |
| NMB | 13% | 102% | | |
| NMGE | 63% | 137% | | |
| $r$ | 0.58 | 0.54 | | |
| COE | -0.30 | -1.84 | | |





**Table 5.** Measured and modelled (addDiesel experiment) OM/OC ratios. Site name abbreviations are given in Table 3.

| Pollutant | Site | Period | Meas. OM/OC | Mod. OM/OC |
|---|---|---|---|---|
| HOA | NK | winter IOP | 1.25 | |
| | NK | summer IOP | 1.19 | |
| | NK | annual | 1.32 | 1.25 |
| | HAR | summer IOP | 1.31 | |
| | DET | winter IOP | 1.45 | |
| SFOA | NK | winter IOP | 1.62 | |
| | NK | annual | 1.78 | 1.70 |
| | DET | winter IOP | 1.64 | |
| SOA | NK | winter IOP | 2.03 | 1.88 |
| | NK | summer IOP | 1.99 | 1.97 |
| | NK | annual | 2.25 | 1.94 |
| | HAR | summer IOP | 2.39 | 1.99 |
| | DET | winter IOP | 2.34 | 1.86 |





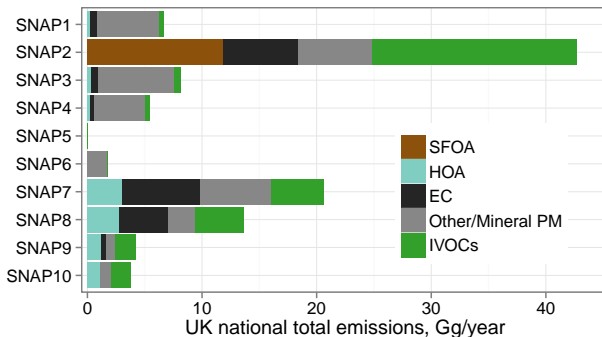

**Figure 1.** Annual UK $PM_{2.5}$ emissions by SNAP sector (Table 1) as specified in the NAEI (for year 2012), with each sector split into POA (HOA or SFOA), EC, and remaining PM following Kuenen et al. (2014). The green bars are additional IVOCs (not included in official emission totals) that can be estimated as 1.5x the POA mass in that sector. They are included in this plot to give an indication of the relative mass of IVOC additions that has been used in other studies.





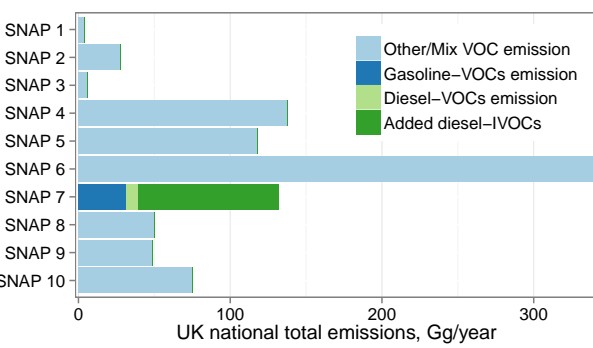

**Figure 2.** Annual UK NMVOC emissions by SNAP sector (Table 1) as specified in the NAEI (for the year 2012), with the SNAP7 emissions sub-divided into gasoline and diesel vehicles, and with the additional diesel-associated IVOC emissions input to the model in this study shown in dark green.

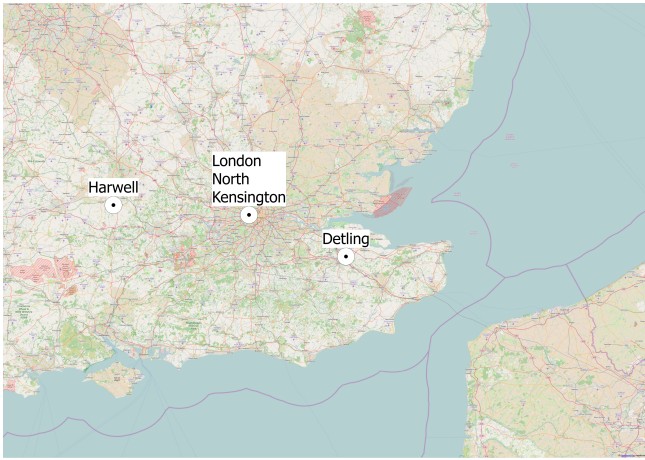

**Figure 3.** Locations of measurement sites used in this study. London North Kensington is an Urban Background site, Harwell and Detling are Rural Background sites. Underlying map from © OpenStreetMap contributors.




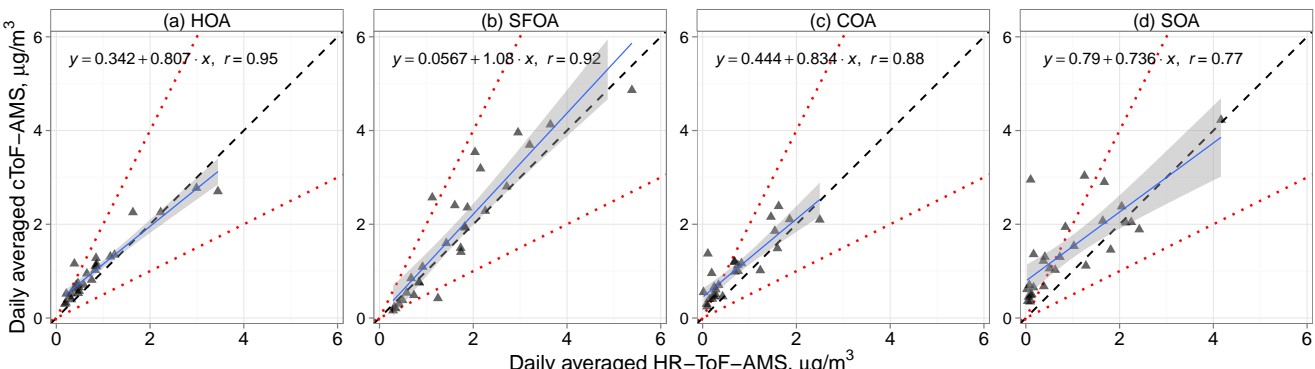

**Figure 4.** Scatterplots of PMF-derived OA component concentrations ((a) HOA, (b) SFOA, (c) COA, (d) SOA) based on different AMS instruments at the London North Kensington site during the winter IOP. The dashed lines are the 2:1, 1:1, and 1:2 lines.

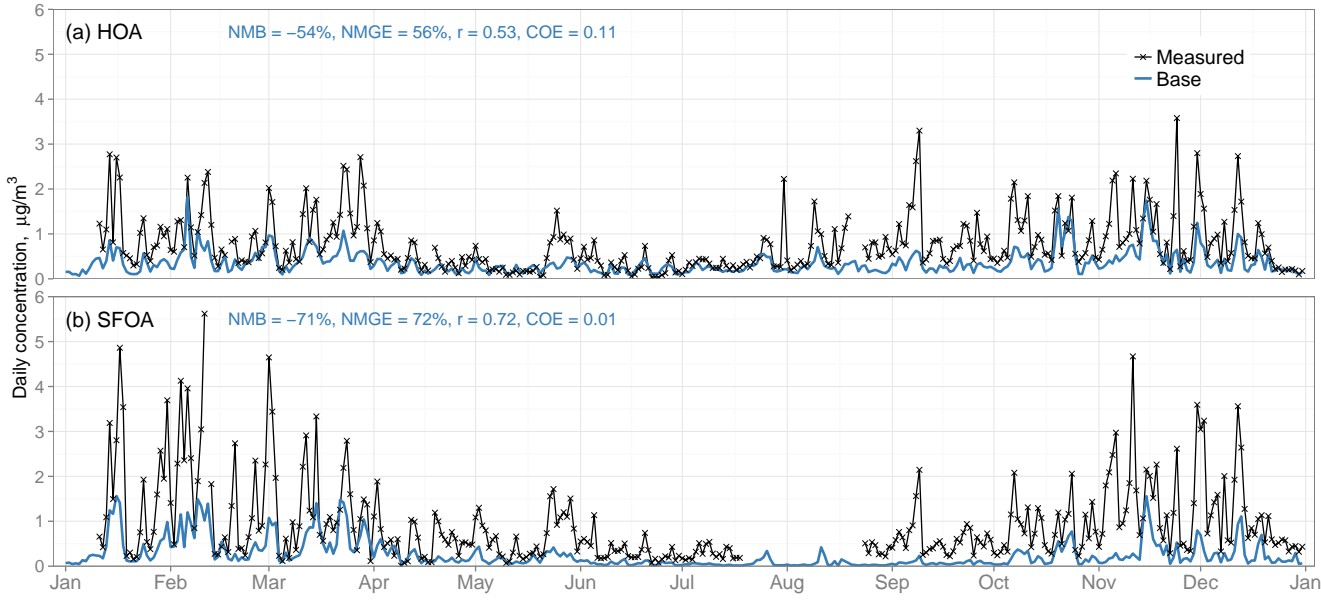

**Figure 5.** Time-series of measured and modelled daily-average concentrations of (a) HOA, and (b) SFOA at the London North Kensington Urban Background site, 2012, measured with the cToF-AMS (Table 3).



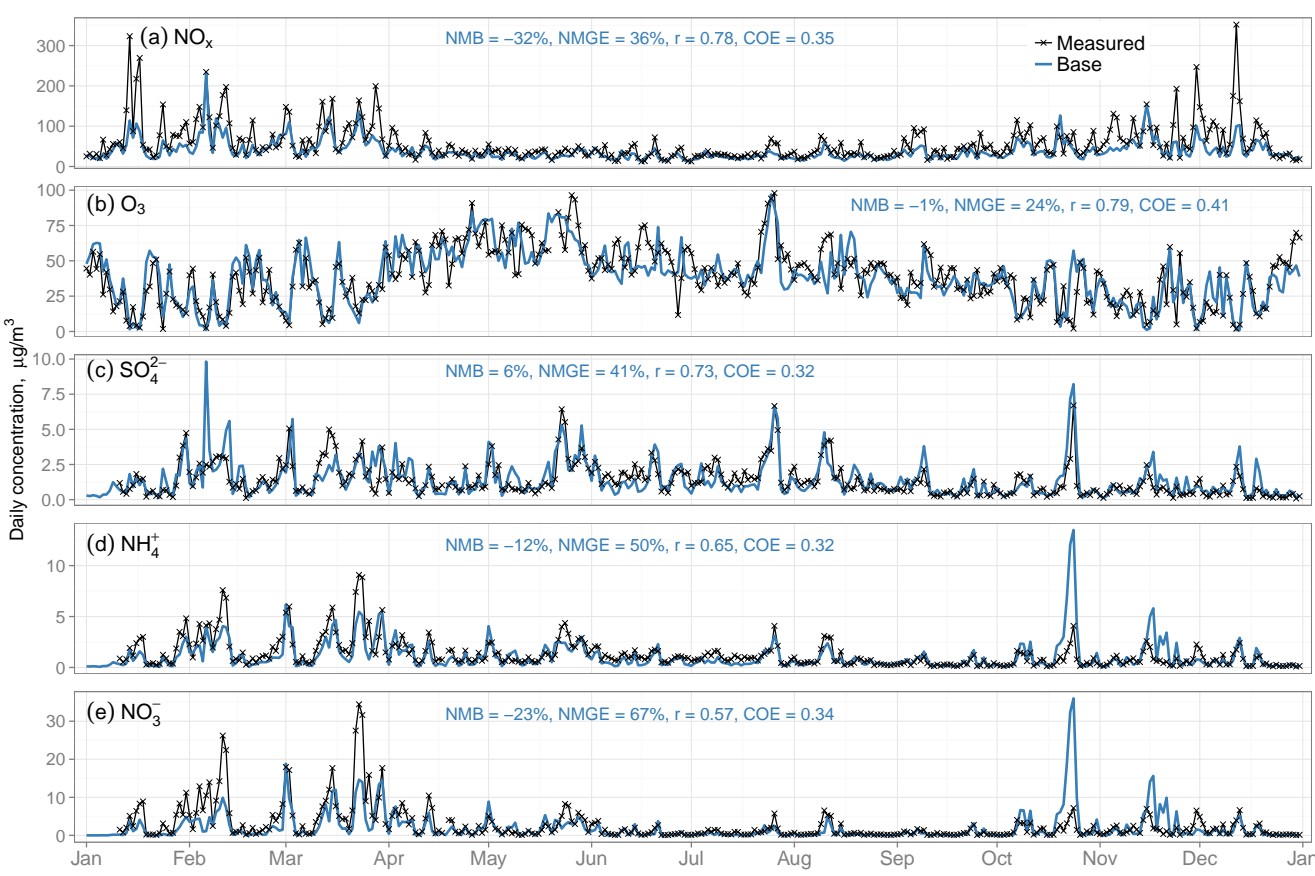

**Figure 6.** Time-series of measured and modelled daily-average concentrations of (a) NO$_x$, (b) O$_3$, (c) SO$_4^{2-}$, (d) NH$_4^+$, and (d) NO$_3^-$ at the London North Kensington Urban Background site, 2012. Measurement data of NO$_x$ and O$_3$ are from the UK Automated Urban and Rural Network (AURN); SO$_4$, NH$_4$ and NO$_3$ were measured with the cToF-AMS (Table 3).





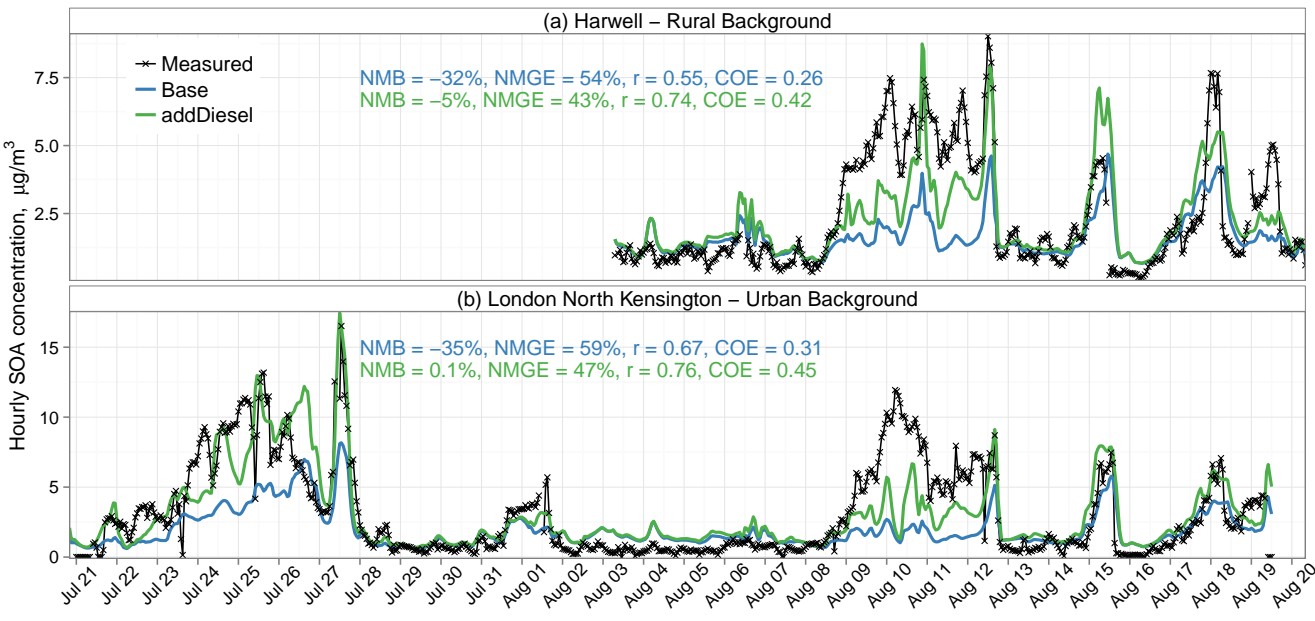

**Figure 7.** Time-series of measured and modelled hourly-average concentrations at (a) the Harwell Rural Background site, and (b) the London North Kensington Urban Background site during the summer IOP. Note the different scales on the y-axes.




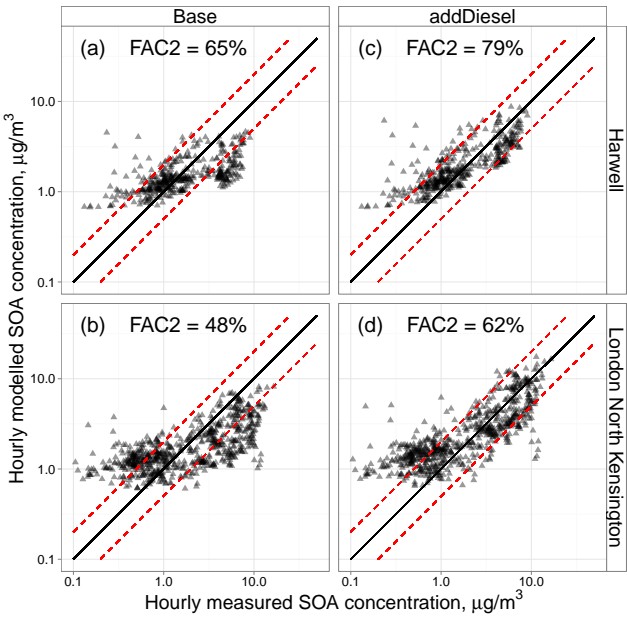

**Figure 8.** Scatterplots of measured and modelled hourly SOA concentrations during the summer 2012 IOP: (a) Base simulation at the Harwell Rural Background site; (b) Base simulation at the North Kensington Urban Background site; (c) addDiesel simulation at the Harwell Rural Background site; (d) addDiesel simulation at the North Kensington Urban Background site. The straight lines are the 2:1, 1:1, and 1:2 lines.

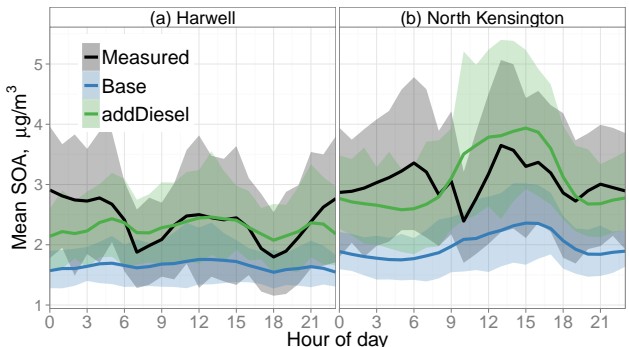

**Figure 9.** Average hourly profiles of modelled and measured SOA during the summer IOP. The shading is the 95% confidence interval.




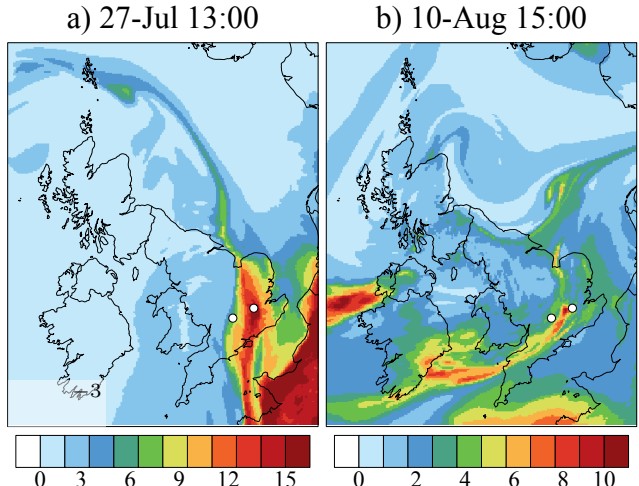

**Figure 10.** Modelled (addDiesel experiment) hourly concentrations of SOA at the time of the maximum measured hourly SOA value at the London North Kensington site during the first and second SOA episodes of the summer IOP. The white circles mark the measurement site locations, left: Harwell, right: London North Kensington.

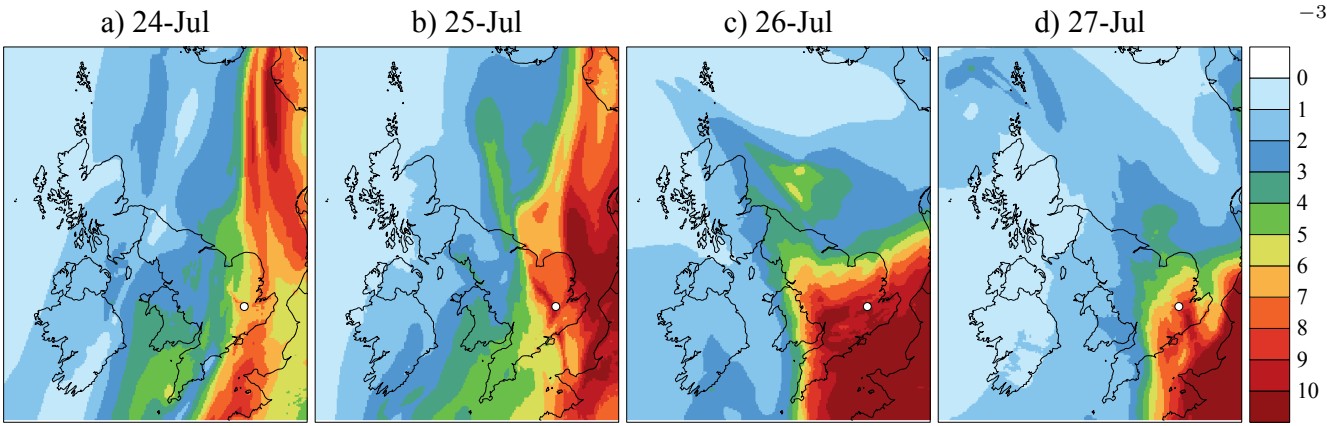

**Figure 11.** Modelled (addDiesel experiment) daily average concentrations of SOA during the first SOA episode of the summer 2012 IOP. The white circle indicates the location of London North Kensington.



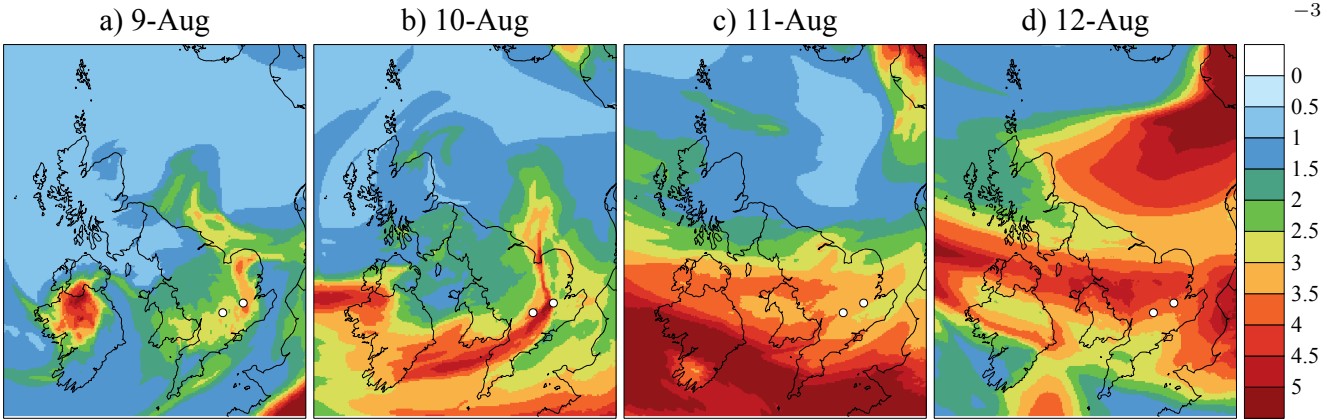

**Figure 12.** Modelled (addDiesel experiment) daily average concentrations of SOA during the second SOA episode of the summer 2012 IOP. The white circles mark the measurement site locations, left: Harwell, right: London North Kensington.

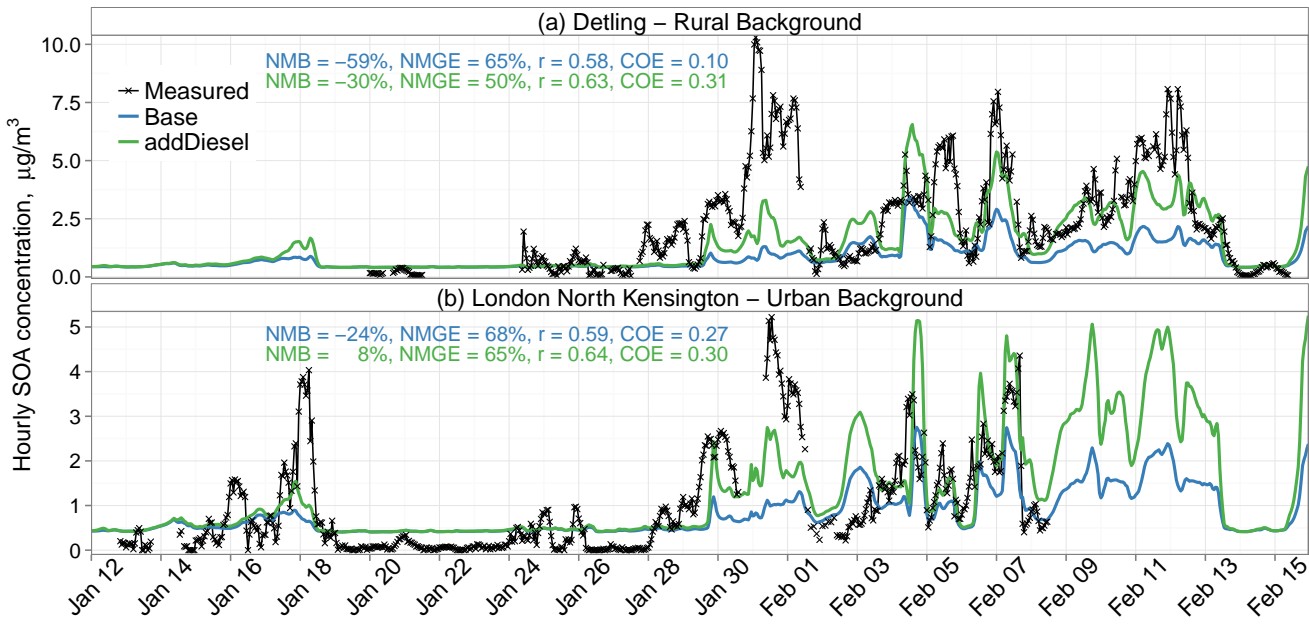

**Figure 13.** Time-series of measured and modelled hourly-average concentrations at (a) the Detling Rural Background site, and (b) the London North Kensington Urban Background site during the winter 2012 IOP. Note the different scales on the y-axes.




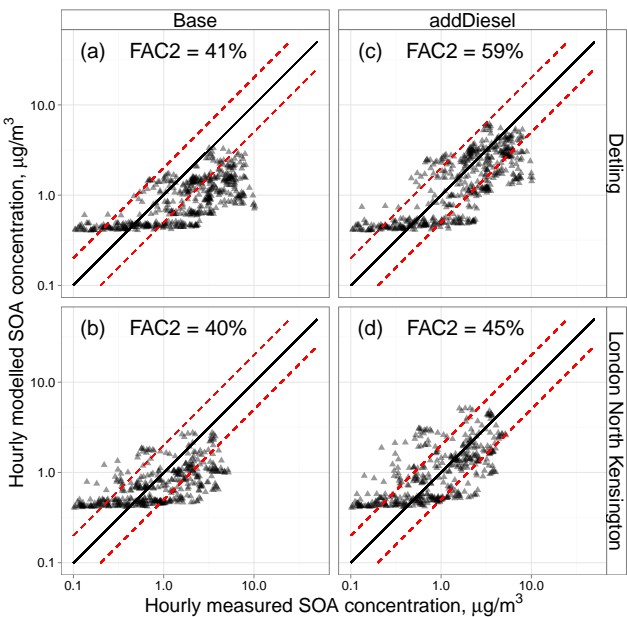

**Figure 14.** Scatterplots of measured and modelled hourly SOA concentrations during the winter 2012 IOP: (a) Base simulation at the Detling Rural Background site; (b) Base simulation at the North Kensington Urban Background site; (c) addDiesel simulation at the Detling Rural Background site; (d) addDiesel simulation at the North Kensington Urban Background site. The straight lines are the 2:1, 1:1, and 1:2 lines.

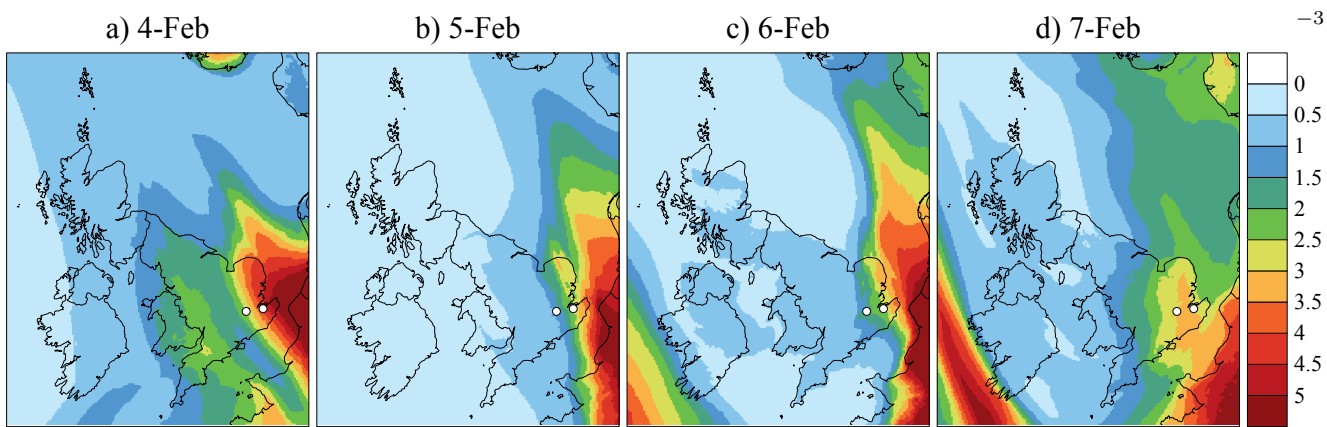

**Figure 15.** Modelled (addDiesel experiment) daily average concentrations of SOA during the second SOA episode of the winter 2012 IOP. The white circles mark the measurement site locations, left: London North Kensington, right: Detling.





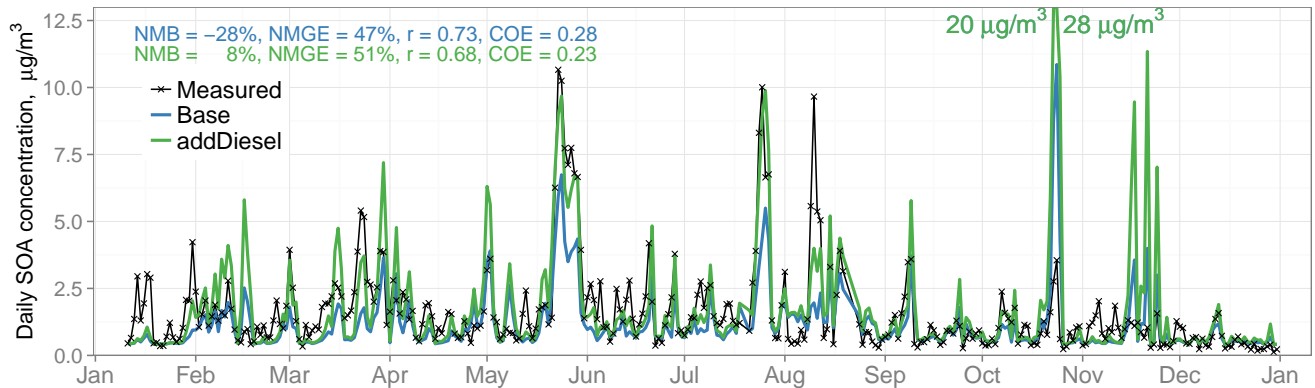

**Figure 16.** Time-series of measured and modelled daily average SOA concentrations at the London North Kensington Urban Background site. The two outliers (23 and 24 October, included in the plot as labels) are excluded from the model evaluation statistics presented in the plot.

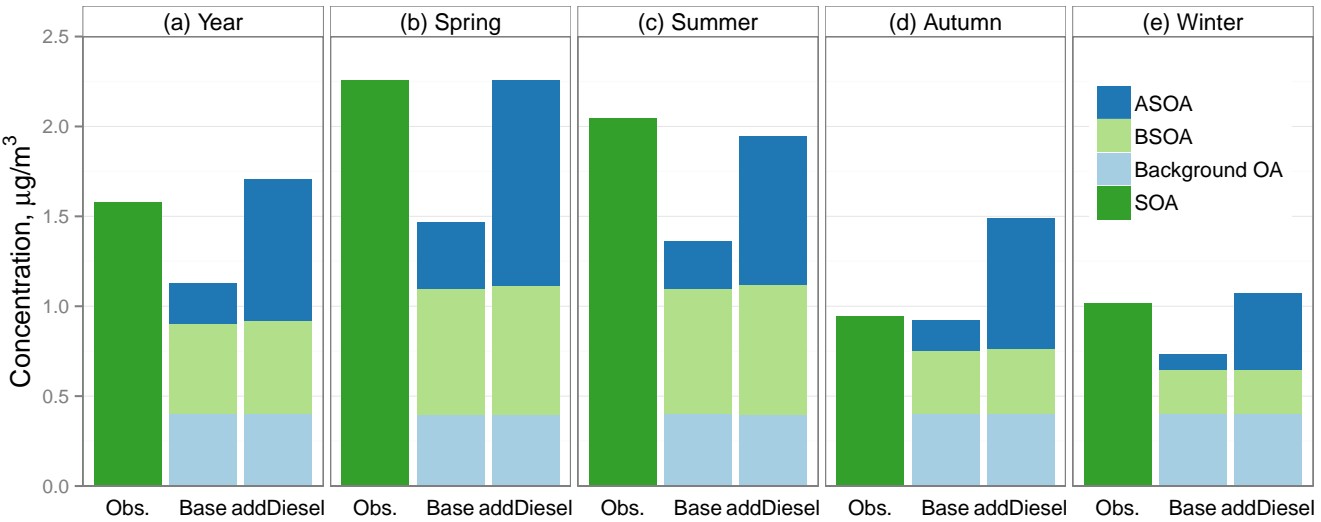

**Figure 17.** Annually and seasonally averaged measured and modelled concentrations of SOA at the London North Kensington site.





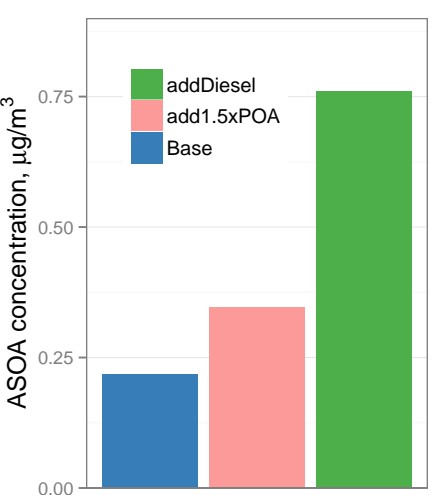

**Figure 18.** Simulated annual and seasonal average concentrations of ASOA for the London North Kensington site of three different model experiments: Base - all emissions as in officially reported emissions inventories; add1.5xPOA - Base + IVOC emissions added proportionally to POA from all source sectors; addDiesel - Base + IVOC emissions from diesel traffic added proportionally to VOC emissions from the on-road traffic source sector (SNAP7); both the latter additions as described in the main text.