# Peer review of "Simulating secondary organic aerosol from missing diesel-related intermediate-volatility organic compound emissions during the Clean Air for London (ClearfLo) campaign"

_Atmospheric Chemistry and Physics, 2015_

## Referee Comment (RC1) · Anonymous Referee #1 · 16 Feb 2016

This is an interesting and generally well-written paper which draws attention to another potentially important source of SOA in the atmosphere. The paper makes nice use of high-quality atmospheric data-sets, and uses not just OA measurements but also other gas and particle data to build confidence in the model. The main conclusions are important: that emissions of IVOC from diesel and their subsequent SOA formation might be much more important than previously assumed.

There are some issues however that I think the authors need to deal with before acceptance for ACP. The main issues I see are:
[Figure]

1. The paper begins (Fig.1, also Sect. 2.5) with an example of a so-called add1.5xPOA approach of illustrating the importance of the 1.5 x PM assumption for IVOC, contrasting with the new addDiesel approach used in this paper (Fig.2). We then see many results from the base and addDiesel cases, but have to wait until almost the very end before seeing some annual-average result from the add1.5xPOA case.

   The 1.5xPM approach used here though differs from that of previous authors, e.g.Robinson et al. (2007), Shrivastava et al. (2008) or even the Bergstrom et al. (2012) paper. In the add1.5xPOA used here, the authors still seem to assume the same inert POA emission as in the base-case, but add IVOC with a very high $C^*$ value. If correct, this is a significant deviation from the other studies, which added SVOC and IVOC across a range of volatilities. If the authors really did just use one IVOC component at $C^*$=1.0e5, then this will lead to more POA close to sources and less downwind compared to e.g. Robinson.

   This begs the question, would a more 'traditional' (Robinson-like) add1.5xPOA scheme give results that might anyway have been better compared to observations than the Base-case used here? This would not mean that diesel-IVOC isn't important, but it might have qualified the relative importance. Of course, one is adding less IVOC and hence producing less SOA. On the other hand, an 1.5xPOA approach (with both SVOC and IVOC) would have generated a bigger gradient between London and the outlying sites, perhaps in better agreement with the observed gradients.

   Concluding, I think they could be better off either (i) re-running with the (dare-I-say) 'traditional' 1.5xPOA for SVOC+IVOC, or (ii) just skipping this test altogether.

2. Given that POA are assumed to be inert, this study likely overestimates OA close to sources. The statement that diesel IVOC can explain about 30% of the annual SOA around London would have to change if POA were allowed to evaporate and react in the atmosphere.

3. The results presented here make a strong case that most SOA is ASOA. This conclusion contrasts strongly with studies based upon radiocarbon and other tracers. Heal et al. (2011) for example suggest a much stronger component from BSOA in Birmingham, and state that this was consistent with other European studies. Can the authors explain this apparent discrepancy?

4. The mass yields for OH oxidation of the n-pentadecane IVOC products is ca. 0.8 for C* up to 10 ug/m3, after correcting for the assumed density, and the great majority of this is one bin, the 10 ug/m3 bin. The potential for much SOA formation is very clear, but I wonder if the authors are exaggerating the amounts. The mass yields are taken from Presto et al. (2010), but that paper suggested that the yields were the product of multi-generational aging, not of the first reaction step. I wonder if aging should have been ignored for these compounds?

Some smaller points follow:

Abstract: I found the first sentence rather vague (what is high-resolution?), and not so interesting (yet another model study). The abstract would make more impact if it began with a comment on the extent of new emissions which forms the basis for this study.

P2, L12. I don't think the results 'prove' that the model has good SOA prediction skill. Even if the comparison with measurements was impressive, there are too many unknowns regarding SOA formation and I don't think any model can claim good skill. I think that this phrase can be omitted.

P3, L64. You need to define the temperature at which these C* values apply.

P3, L76. Why mention AMS for organic PM? I don't think many European PM inventories make use of AMS data.

P4, L114. Define PMF. Also, which PMF method was used?

P5, L132. WRF can be set up in many different ways, with varying impacts on accuracy for air pollutant applications. Please give more details or a suitable reference.

P5, L137, specify anthropogenic emissions here.

P5, L138, specify aerodynamic diameter.

P5, L142. Why use a paragraph on an NFR system which is not used in this work? Delete.

P6, L156. The term SFOA is confusing, and wasn't used by Bergstrom et al as claimed here. If I understand right SFOA includes biomass burning (which is usually said to give BBOA), but also coal and charcoal.

P6, L173. It could be noted that the Jathar et al. (2014) study also suggested different ratios of IVOC to PM than those of Shrivastava et al.

P7, L191. 'under modeled SOA' - do you mean when comparing with observations?

P7, L200. Shouldn't you also mention aromatics and other compounds here.

P7, L207. Which GC x GC system? Explain what is meant.

P7, L209. Any reference for the number of studies providing that rate constant?

p8, L247. Biogenic emissions of what?

p9. Sect. 2.6 'Comparison with measurements': This section can be simply renamed 'Measurements, since that is what it deals with down to L288.

P9, L259. Why have references to Fig. 3? Give the references after the mention of each site, or add 'site details given in' or some such phrase.

P9, L274. Are you sure that European inventories don't include cooking OA? I think it may be underestimated, but am not sure it is ignored completely.

P9, L276. This sentence was confusing. I can see that two instruments can disagree,

but what does it mean if there is just one instrument? Can an AMS and its PMF disagree, or what?

P9, L284 .... what period/site/analysis are these sentences and statistics referring to?

P9, L289 on. This small section on statistical metrics has nothing to do with the discussion of AMS etc which it follows, and could be set in a small section of its own.

P9, L291. I don't think correlation coefficient needs a reference to Carslaw and Ropkins; 'r' has been used for many many years before that paper was written. Actually, NMGE might need more explanation. All these could usefully be defined in supplementary.

P10, equation (1): explain what $i$ and $n$ are.

P10, L298. Re-phrase - it sounds as though the measurement mean is better at capturing the variation in measurements than the model.

P10, L303. This bit about WRF could be moved to Sect. 2.1.

P11, around L35. All these numbers for NMB, etc. could be tabulated for easier comparison.

P12, L364. This refers to Fig. 12a,b, but there are no a,b labels in Fig.12

P12, L389. Re-phrase (or omit). It is obvious from the plots that this background OA is an overestimate for some days at least.

P13, L405 on. The whole discussion here is in terms of SOA and IVOC. But, how did the model perform for NOx and CO for these 'difficult' periods - maybe the problem is dispersion rather than IVOC? Or maybe the model's enthalpy values are wrong, and don't respond to cold temperatures as they should. I don't see why problems are blamed on domestic sources either. Wouldn't for example cold-starts for vehicles also produce more POA/IVOC, or commercial premises use more fuel in cold conditions? Are wood-burning emissions really an issue in London?

P15, L475. The title says comparison to previous (IVOC=1.5xPOA) approach, but as noted above, the method used here seems to be unique; not that of earlier papers.

P15-P16. The authors make various policy recommendations, e.g. (P15, L500) 're-finements should be reported to CEIP' and a very specific recommendation for PM1, PM1-2.5 and PM2.5-10 on P16, L533. Why not just suggest submission of size-distributions? Why no mention of volatility - the VBS approach almost begs for people to submit emissions in different volatility classes. And since this paper is really exploring IVOC and not PM emissions per se, why didnt the authors focus on those?

Actually, I suggest that the authors don't try to tell countries what to do, but rather discuss any scientific insights into emission reporting that this study on IVOC reveals.

P16, L509. Can't small changes in SOA (or any pollutant) also be a reflection of long-range transport? Not all pollution is formed at short time-scales close to source.

P16, L520. Where did the value 4.0e(-12) come from for ASOA and BSOA oxidation?

P16, L520 on. This section offers a few 'tuning' suggestions, but there are always any number of these in the field of SOA formation. For example, recent studies have suggested that SOA formation should be much greater than previously assumed, perhaps by a factor of four or so (Zhang et al., 2014).

P16, L527. I would say that this paper illustrates the potential for a significant contribution, rather than that they can quantify the relative impact. To do the latter, one would need to be sure that all relative impacts are reasonably well known, and that clearly isn't the case.

P17, Sect.5. The first paragraph simply repeats sentences from Sect. Don't do that.

This section should also mention the results for other pollutants (NOx, O3, etc.), which are the main reason one can have some confidence in the basic modeling system. (Use of such data is one of the strengths of this paper I think.)

P17, L563. imported from where?

P17, last paragraph. The interpretation of what contributes to the 90-th percentile is not so easy I think. And I certainly don't think one can state that 40% is due to missing diesel precursors. SOA is too complex for such simple statements.

P20, L635. Expand/explain CEIP. Is this the name of a report, or just a web side?

P21, L665, EEA, Entec - these references are too short for readers to understand or find. Give proper references, with addresses as necessary.

P29, Fig. 1. The caption should state early that this is PM25 *and* SVOC/IVOC gases. (The issue of IVOC or SVOC+IVOC is discussed above, but it complicates this figure.)

P31. Fig. 5. Explain 'Base' as used in legend.

P31. Fig. 6(a). NOx is a the sum of both NO and NO2. Were they really summed with own molecular weights, or is this as NO2? (ppb would have been easier to interpret!). Also, in the captions, add the superscript ion-labels too.

P33, Fig. 9. This Figure looks very fine on-screen, but when printing out it looks very different - much of the black seems to be over-written with green and/or blue. Please use a different figure format, and check the printout.

P34, Figs. 11-12. We don't really need these since Fig. 10 has made the point about gradients well. Move to Supplementary.

P35, Fig. 14. Why use log-log plots? Wouldn't a simple liner plot better display the range of data?

P36, Fig. 16. Here the addDiesel statistics look worse than the base-results. Are these really consistent with data presented in Table 4?

Figs and color schemes. The colors used for ASOA, addDiesel, etc seem to vary randomly from figure to figure (e.g. green is addDiesel in Fig.9 but observed SOA in Fig. 17. Please harmonize.

ADDITIONAL REFERENCES:

Heal, MR et al. Application of 14C analyses to source apportionment of carbonaceous PM2.5 in the UK, Atmos. Environ., 45, 2341-2348, 2011

Zhang, X., et al., Influence of vapor wall loss in laboratory chambers on yields of secondary organic aerosol, PNAS, 111, no. 16, 5802-5807, 2014

---

## Referee Comment (RC2) · Anonymous Referee #2 · 20 Feb 2016

In this paper, Ots et al. present an interesting method to account for the emissions of intermediate volatile organic compounds (IVOCs). They suggest that VOC emissions can be added proportionally to VOC emissions as opposed to the POA emissions which is the standard method used by current Volatility Basis Set (VBS) models. This approach can potentially pave the way for an accurate representation of IVOCs in the emission inventories which is proved to be a necessity for SOA models during the last decade. Overall, the manuscript is well written and scientifically sound. I recommend this study for publication after taking the following comments into account.

[Figure]

General comment:

1. The authors include additional diesel related IVOC emissions based on the VOC emissions from the transport sector. This resulted in a significant improvement of their model results which brought the predicted SOA close to measurements during winter, spring, and summer while it resulted in an overprediction during autumn (Fig. 17 of the manuscript). However, the transport sector consist only one of the ten sectors that their emission inventory includes. This raise the question of how much their model performance will change (towards overprediction) if they will add the missing IVOC emissions from the rest nine sectors. Do they have indications that the only important source of IVOCs is the transport sector? While I strongly support the suggested approach of deriving the IVOC emissions based on intermediate length alkanes (or naphthalene seen in other studies; Pye and Seinfeld, 2010) I am quite sceptic about the impact shown here by only one sector. I suggest adding a discussion on this matter, probably in section 4.

Specific comments

1. Page 2 line 20: Biomass burning OA (BBOA) is also a usual component that PMF can identify. Does SFOA correspond to BBOA? If so, you should use the latter since it is more commonly used by the AMS community. Furthermore, you should also report the oxygenated organic aerosol (OOA) which then can be split into LV-OOA and SV-OOA.

2. Page 2 lines 22-27: Since the main focus of the manuscript is the simulation of SOA, it would be good to add a sentence regarding the performance of the global models in terms of SOA (e.g.,Spracklen et al., 2011;Jathar et al., 2011;Jo et al., 2013;Mahmud and Barsanti, 2013;Shrivastava et al., 2015;Tsimpidi et al., 2016)

3. Page 3 line 20: Please add recently developed models that follow the same assumption in order to indicate that the factor of 1.5 is widely used up to date (e.g.,Koo et al., 2014;Tsimpidi et al., 2014)

4. Page 4: The line numbers here and in a number of the following pages are not correct. They should either restart in each page or continue throughout the text.

5. Page 5 line 13: Do HOA and SFOA correspond to the fossil fuel combustion and domestic combustion of you emission inventory (EI)? Please clarify since it is not clear if you used these fractions to convert the OC from your EI to OA.

6. Pages 5 line 28: Assuming that POA is treated as non-volatile will result in unrealistically high OA concentrations in the aerosol phase. This will favor the partitioning of your semivolatile compounds (e.g. oxidation products of IVOCs) into the aerosol phase resulting in an overestimation of SOA as well. On the contrary, if you do not assume that POA and SOA participate in the same solution during the phase partitioning, you should expect an underestimation of SOA. Please comment at this point on the implications of your assumption regarding the POA volatility.

7. Page 6 line 2: Tsimpidi et al. (2010) used 4 volatility bins to distribute the oxidation products of VOCs. Can you please report the aerosol yields for the 5th volatility bin that you are using ($C^*=0.1$) and add a reference for them as well?

8. Page 6 line 6: According to Lane et al. (2008) the use of aging reactions improved their results compared to measurements from urban areas but resulted in a strong overprediction over rural areas. They attributed this discrepancy to a potential balancing of decomposition to smaller more volatile products (fragmentation) and production of more substituted less volatile products (functionalization) during the photochemical aging of biogenic SOA. This was also confirmed by laboratory studies (Ng et al., 2006). Therefore they suggested that no ageing of biogenic SOA should be considered. Furthermore, the use of an ageing rate constant of $4.0 \times 10^{-12}$ cm3 molecule-1 s-1 is kind of conservative compared to what is used lately by most models (i.e. $1.0 \times 10^{-11}$ cm3 molecule-1 s-1; Fountoukis et al. 2014). According to the above, I suggest either changing your scenarios by using $1.0 \times 10^{-11}$ cm3 molecule-1 s-1 as a rate constant and assume no ageing of biogenic compounds or to make a sensitivity test and investigate the effect of these assumptions on your results (especially regarding the rate constant).

9. Page 6 line 10: What is the saturation concentration of this "background OA" compound? Is this considered nonvolatile since it is very aged and highly oxygenated? Furthermore, please provide a reference for assigning a value of 0.4 $\mu$g m-3 to this compound. Does this value based on measurements?

10. Page 7 line 35 (on the top of the page): What is the SOA mass yield that you used for the 1000 $\mu$g m-3 volatility bin? Please provide a reference as well.

11. Page 7 lines 28-29: A more fair comparison between the two approaches would be to add IVOCs proportionally to POA from sector 7 only as you did on your addDiesel scenario. Can you investigate this additional scenario as well?

12. Page 9 Section 3.1: Since POA are assumed to be nonvolatile you would expect to overpredict their concentrations. Are the emissions so severely underestimated? Please report that the presented underprediction will be even more significant if you add the semivolatile character of POA.

13. Page 9 line 20: Please replace the "secondary pollutants" with "secondary inorganic pollutants"

14. Page 10 lines 7-8: Add a reference to Fig. 7

15. Page 13 line 30 (on the top of the page): How you calculated the 40%?

16. Page 15 lines 4-5 (or 33-34): Pye and Seinfeld (2010) have used a naphthalene-like surrogate specie to describe IVOCs instead of the traditional "POA" method. Please refer to this work as well (maybe in the introduction).

References

Fountoukis, C., Megaritis, A. G., Skyllakou, K., Charalampidis, P. E., Pilinis, C., van der Gon, H., Crippa, M., Canonaco, F., Mohr, C., Prevot, A. S. H., Allan, J. D., Poulain,

[Figure]

L., Petaja, T., Tiitta, P., Carbone, S., Kiendler-Scharr, A., Nemitz, E., O'Dowd, C., Swietlicki, E., and Pandis, S. N.: Organic aerosol concentration and composition over Europe: insights from comparison of regional model predictions with aerosol mass spectrometer factor analysis, Atmospheric Chemistry and Physics, 14, 9061-9076, 10.5194/acp-14-9061-2014, 2014.

Jathar, S. H., Farina, S. C., Robinson, A. L., and Adams, P. J.: The influence of semivolatile and reactive primary emissions on the abundance and properties of global organic aerosol, Atmos. Chem. Phys., 11, 7727-7746, 2011.

Jo, D. S., Park, R. J., Kim, M. J., and Spracklen, D. V.: Effects of chemical aging on global secondary organic aerosol using the volatility basis set approach, Atmos. Environ., 81, 230-244, 2013.

Koo, B., Knipping, E., and Yarwood, G.: 1.5-Dimensional volatility basis set approach for modeling organic aerosol in CAMx and CMAQ, Atmospheric Environment, 95, 158-164, 10.1016/j.atmosenv.2014.06.031, 2014.

Mahmud, A., and Barsanti, K.: Improving the representation of secondary organic aerosol (SOA) in the MOZART-4 global chemical transport model, Geoscientific Model Development, 6, 961-980, 10.5194/gmd-6-961-2013, 2013.

Ng, N. L., Kroll, J. H., Keywood, M. D., Bahreini, R., Varutbangkul, V., Flagan, R. C., Seinfeld, J. H., Lee, A., and Goldstein, A. H.: Contribution of first- versus second-generation products to secondary organic aerosols formed in the oxidation of biogenic hydrocarbons, Environ. Sci. Technol., 40, 2283-2297, 2006.

Pye, H. O. T., and Seinfeld, J. H.: A global perspective on aerosol from low-volatility organic compounds, Atmos. Chem. Phys., 10, 4377-4401, 2010.

Shrivastava, M., Easter, R. C., Liu, X., Zelenyuk, A., Singh, B., Zhang, K., Ma, P.-L., Chand, D., Ghan, S., Jimenez, J. L., Zhang, Q., Fast, J., Rasch, P. J., and Tiitta, P.: Global transformation and fate of SOA: Implications of low-volatility SOA and gasphase fragmentation reactions, Journal of Geophysical Research-Atmospheres, 120, 4169-4195, 10.1002/2014jd022563, 2015.

Spracklen, D. V., Jimenez, J. L., Carslaw, K. S., Worsnop, D. R., Evans, M. J., Mann, G. W., Zhang, Q., Canagaratna, M. R., Allan, J., Coe, H., McFiggans, G., Rap, A., and Forster, P.: Aerosol mass spectrometer constraint on the global secondary organic aerosol budget, Atmospheric Chemistry and Physics, 11, 12109-12136, 10.5194/acp-11-12109-2011, 2011.

Tsimpidi, A. P., Karydis, V. A., Pozzer, A., Pandis, S. N., and Lelieveld, J.: ORACLE (v1.0): module to simulate the organic aerosol composition and evolution in the atmosphere, Geoscientific Model Development, 7, 3153-3172, 10.5194/gmd-7-3153-2014, 2014.

Tsimpidi, A. P., Karydis, V. A., Pandis, S. N., and Lelieveld, J.: Global combustion sources of organic aerosols: Model comparison with 84 AMS factor analysis data sets, Atmos. Chem. Phys. Discuss., 2016, 1-51, 10.5194/acp-2015-989, 2016.

---

## Author Comment (AC1) · 5 Apr 2016

**acp-2015-920: Simulating secondary organic aerosol from missing diesel-related intermediate-volatility organic compound emissions during the Clean Air for London (ClearfLo) campaign**

**Response to Reviewer 1 comments**

We thank this reviewer for their supportive and helpful comments. Below we respond to each comment individually.

We note that this reviewer has used line numbers from the version of the manuscript submitted for the technical review, rather than the version of the manuscript openly published in the Discussion phase.

The reviewer's comments are in italics and blue font, our responses are in normal text.

*This is an interesting and generally well-written paper which draws attention to another potentially important source of SOA in the atmosphere. The paper makes nice use of high-quality atmospheric data-sets, and uses not just OA measurements but also other gas and particle data to build confidence in the model. The main conclusions are important: that emissions of IVOC from diesel and their subsequent SOA formation might be much more important than previously assumed.*
*There are some issues however that I think the authors need to deal with before acceptance for ACP. The main issues I see are:*

> *1. The paper begins (Fig.1, also Sect. 2.5) with an example of a so-called add1.5xPOA approach of illustrating the importance of the 1.5 x PM assumption for IVOC, contrasting with the new addDiesel approach used in this paper (Fig.2). We then see many results from the base and addDiesel cases, but have to wait until almost the very end before seeing some annual-average result from the add1.5xPOA case.*
> *The 1.5xPM approach used here though differs from that of previous authors, e.g.Robinson et al. (2007), Shrivastava et al. (2008) or even the Bergstrom et al. (2012) paper. In the add1.5xPOA used here, the authors still seem to assume the same inert POA emission as in the base-case, but add IVOC with a very high C* value. If correct, this is a significant deviation from the other studies, which added SVOC and IVOC across a range of volatilities. If the authors really did just use one IVOC component at C\*=1.0e5, then this will lead to more POA close to sources and less downwind compared to e.g. Robinson.*
> *This begs the question, would a more 'traditional' (Robinson-like) add1.5xPOA scheme give results that might anyway have been better compared to observations than the Base-case used here? This would not mean that diesel-IVOC isn't important, but it might have qualified the relative importance. Of course, one is adding less IVOC and hence producing less SOA. On the other hand, an 1.5xPOA approach (with both SVOC and IVOC) would have generated a bigger gradient between London and the outlying sites, perhaps in better agreement with the observed gradients.*
> *Concluding, I think they could be better off either (i) re-running with the (dare-I-say) 'traditional' 1.5xPOA for SVOC+IVOC, or (ii) just skipping this test altogether.*

Response: We have now undertaken the reviewer's suggestion (i) and run a model experiment identical to Shrivastava et al. 2008. We have amended text accordingly in both the Methods and in the Results (Sect. 3.6, Fig. 18). The relevant text in the Methods now reads:

"Nevertheless, we have also performed a model run using the POA-based IVOC estimate, also including the semivolatile treatment of POA. The emitted semivolatile POA (SVOCs) and 1.5xPOA IVOCs are assigned to 9 VBS bins: 0.03xPOA, 0.06xPOA, 0.09xPOA, 0.14xPOA, 0.18xPOA, 0.30xPOA, 0.40xPOA, 0.50xPOA, 0.80xPOA to the bins $0.01$–$10^6$, respectively; totalling 2.5xPOA (Shrivastava et al. 2008). Both SVOCs and IVOCs then go through atmospheric ageing with OH ($k = 4.0 \times 10^{-11}$ $cm^3$ molecule$^{-1}$ s$^{-1}$; Shrivastava et al. (2008)). /.../ SVOCs and IVOCs that have undergone at least one ageing shift and are in the particulate phase are included under SOA (in addition to ASOA and BSOA from VOCs as in the Base case)."

And Sect. 3.6 now reads:
**"3.6 Comparison to the previous (IVOCs=1.5xPOA) approach**
Figure 18 shows the annual average HOA, SFOA, BSOA and Background OA (BGND OA), and ASOA concentrations at London North Kensington modelled with different assumptions for additional IVOC emissions. As was explained in Sect. 2.4, for the UK, the addDiesel experiment adds 90 Gg of diesel-related IVOCs proportionally to road transport emissions (SNAP7), whereas the IVOCs=1.5xPOA approach only adds 5 Gg to SNAP7 and another 26 Gg to other sectors (mainly to SNAP2: residential and non-industrial combustion). Therefore, our approach creates a considerably larger amount of SOA from IVOCs (and only from diesel-related IVOCs) than the previous method. The 1.5volPOA experiment was undertaken using the semivolatile treatment of POA emissions. This means that the modelled ASOA from this experiment also includes aged semivolatile POA, possibly giving it potential to create more ASOA than the Base or addDiesel experiments (the organic material added to the model in the 1.5volPOA experiment is 1.0×POA (as SVOCs) + 1.5×POA (IVOCs) = 2.5×POA as introduced by Robinson et al. (2007) and Shrivastava et al. (2008)). It can be seen from Figs. 18a, b that treating POA as semivolatile leads to much lower concentrations than the nonvolatile treatment (which already underestimates measured concentrations of HOA and SFOA by -54% and -71%, respectively; Fig.5). This is not surprising since with the semivolatile treatment of POA only 3% + 6% + 9% of the POA is assigned to the three lowest volatility bins with saturation concentrations of 0.01, 0.1 and 1 $\mu g\,m^{-3}$ (as given in Sect. 2.4). In a study in Mexico City, Shrivastava et al. (2011) then revised this treatment – assuming much higher total semi- and intermediate volatility POA emissions: 7.5 × the inventory emissions of (particulate) POA. This was justified by the fact that their emission factors of POA were derived from measurements at urban background sites, but, following Robinson et al. (2007), 2/3 of POA would have evaporated by then. (Recently, Shrivastava et al. (2015) also used this factor of 7.5 in global simulations.) Emission factors used in European inventories are, however, taken from tailpipe measurements with concentrations sufficiently high that most of the semivolatiles should still be reported in the particulate phase. Therefore the further underestimation of HOA and SFOA concentrations with the volatile treatment could be due to a number of issues: (i) a systematic underestimation of emissions, but for a different reason than in Shrivastava et al. (2011), (ii) the volatility of POA is overestimated by Robinson et al.

(2007), (iii) the evaporation of semivolatile POA emission is too rapid in the model (instantaneous in our set-up).

Figure 18c shows that the lower HOA and SFOA concentrations lead to a very small negative change for the absorptive partitioning of BSOA. Finally, it can be seen from the annual average concentrations of ASOA in Fig. 18d that including aged SVOCs and IVOCs in the simulation doubles the modelled ASOA concentration compared to the Base case scenario (ASOA from officially reported anthropogenic VOCs), but that the ASOA in our 1.5volPOA experiment is still much lower than simulated with our addDiesel experiment."

[Figure]

**Figure 18.** Simulated annual and seasonal average concentrations of OA components (BGND OA stands for Background OA) for the London North Kensington site of three different model experiments: Base - all emissions as in officially reported emissions inventories, POA is treated as non-volatile; 1.5volPOA - semivolatile treatment of POA + IVOC emissions added as 1.5xPOA; addDiesel - Base + IVOC emissions from diesel traffic added proportionally to VOC emissions from the on-road traffic source sector (SNAP7); both the latter additions as described in the main text.

(Figure numbering as in the revised manuscript.)

*2. Given that POA are assumed to be inert, this study likely overestimates OA close to sources. The statement that diesel IVOC can explain about 30% of the annual SOA around London would have to change if POA were allowed to evaporate and react in the atmosphere.*

Response: We derive the "30%" from the change in normalised mean bias (so in comparison to the measurements), not in comparison to total modelled SOA, and therefore we believe

this is an appropriate value for our statement highlighting the potential contribution of these additional emissions.

> *3. The results presented here make a strong case that most SOA is ASOA. This conclusion contrasts strongly with studies based upon radiocarbon and other tracers. Heal et al. (2011) for example suggest a much stronger component from BSOA in Birmingham, and state that this was consistent with other European studies. Can the authors explain this apparent discrepancy?*

Response: It is certainly not our intention to make a case that most SOA in London was ASOA, and we do not believe that our text does imply this claim. From Fig.17a in our manuscript (barplot of modelled annual average ASOA/BSOA/Background OA concentrations), it can be calculated that the ratio of modelled BSOA+Background OA to total SOA is 53%. Therefore even with the additional ASOA generated from the additional IVOC emissions, about half of the simulated SOA is still of biogenic origin, which is therefore not inconsistent with the experimental measurements using the radioisotope of carbon (14C) reported in Heal et al. (2011). It needs also to be noted that whilst 14C is an ideal tracer for distinguishing between fossil and contemporary carbon, it cannot directly distinguish whether the carbon is of primary or secondary origin. Therefore, for a direct comparison with a modelling study like ours, all primary components (such as the HOA and SFOA) would also have to be well represented in the model, but we showed a -71% bias for SFOA. This bias is caused by the fact that the national atmospheric emissions inventory assumes zero domestic wood and coal burning emissions in London, as it is a smoke control area (and therefore residential burning of these solid fuels is not allowed), but recent studies (e.g. Crilley et al. 2015) have concluded that there are indeed local sources of SFOA in London. Furthermore, in the current set-up, we model SFOA as one entity (so wood and coal together), but in the 14C analysis, OA from wood-burning and coal would be apportioned into different categories (contemporary and fossil, respectively).

To emphasise the above point, we have now added the following sentence to the end of Section 3.4: "We note that Fig. 17a shows that in the addDiesel simulation, the modelled BSOA+Background OA still makes up 53% of the SOA, as an annual average."

> *4. The mass yields for OH oxidation of the n-pentadecane IVOC products is ca. 0.8 for C\* up to 10 ug/m3, after correcting for the assumed density, and the great majority of this is one bin, the 10 ug/m3 bin. The potential for much SOA formation is very clear, but I wonder if the authors are exaggerating the amounts. The mass yields are taken from Presto et al. (2010), but that paper suggested that the yields were the product of multi-generational aging, not of the first reaction step. I wonder if aging should have been ignored for these compounds?*

Response: Our view is that ageing should not be ignored for these compounds. Although Presto et al. (2010) report that these yields almost certainly include multigenerational processing during the experiment, they also conclude (in the last paragraph of their paper): "The slow rise in f44 in experimental systems - a few percent over several hours - further indicates that OA constantly evolves over long time scales, on the order of days, and that short chamber experiments likely do not reproduce the complete transformation from emissions to OOA."

We do acknowledge, however, that the ageing rates and assumptions used in VBS modelling studies can vary quite widely between different studies, as is noted in the Discussion (paragraph beginning "We use an ageing rate of…").

**Responses to smaller points:**

1. *Abstract: I found the first sentence rather vague (what is high-resolution?), and not so interesting (yet another model study). The abstract would make more impact if it began with a comment on the extent of new emissions which forms the basis for this study.*

Response: We do not believe the opening sentence is vague as it provides a number of specific contextual facts to the work; namely, that it is an ACTM modelling study, that the atmospheric process of interest is SOA formation, that the geographical context is the UK, and that the period of study is the full year 2012. However, we accept that the phrasing "high resolution" is not specific and that we could introduce the concept of new estimates of emissions into the opening sentence (although we note the latter is encapsulated in the title of the paper). We have therefore amended the opening sentence to now read: "We present high-resolution (5 km × 5 km) atmospheric chemistry transport model (ACTM) simulations of the impact of new estimates of traffic-related emissions on secondary organic aerosol (SOA) formation over the UK for 2012."

2. *P2, L12. I don't think the results 'prove' that the model has good SOA prediction skill. Even if the comparison with measurements was impressive, there are too many unknowns regarding SOA formation and I don't think any model can claim good skill. I think that this phrase can be omitted.*

Response: We agree that the text referring to good model skill is not appropriate for the abstract and have now omitted this sentence.

3. *P3, L64. You need to define the temperature at which these C* values apply.*

Response: The temperature of 298 K has now been added.

4. *P3, L76. Why mention AMS for organic PM? I don't think many European PM inventories make use of AMS data.*

Response: It was not our intention to imply that national inventories use AMS data. However, on review we note that our original text referring to AMS measurements of the OC content of particles is not relevant to the point being made in this sentence that SVOC and IVOC species are hard to measure, so we have now deleted it. The modified text now reads: "Current emissions inventories, however, only report estimates for VOCs and for the particle fraction of the emissions of species with lower volatilities. The main reason for this is that compounds with intermediate volatility (SVOCs and IVOCs) are difficult to quantify and this is currently not routinely undertaken."

5. *P4, L114. Define PMF. Also, which PMF method was used?*

Response: The acronym PMF is defined already in the second of paragraph of the Introduction. We have amended a sentence in Sect. 2.6 to include the PMF methods:

"A detailed description of the derivation and optimization of the factors retrieved from the AMS data at Detling can be found in Xu et al. (2016), at London North Kensington in Young et al. (2015a) and Young et al. (2015b) (all three of these analyses were performed with the PMF2 solver), and at Harwell in Di Marco et al. (2015) (using the ME-2 solver)."

6. *P5, L132. WRF can be set up in many different ways, with varying impacts on accuracy for air pollutant applications. Please give more details or a suitable reference.*

Response: We have added the following sentence: "The WRF configuration was as follows: Lin Purdue for microphysics; Grell-3 for cumulus parametrization; Goddart Shortwave for radiation physics; and Yonsey University (YSU) for planetary boundary layer (PBL) height (see NCAR (2008) for further information)."

7. *P5, L137, specify anthropogenic emissions here.*

Response: Done.

8. *P5, L138, specify aerodynamic diameter.*

Response: Done.

9. *P5, L142. Why use a paragraph on an NFR system which is not used in this work? Delete.*

Response: The text is now deleted.

10. *P6, L156. The term SFOA is confusing, and wasn't used by Bergstrom et al as claimed here. If I understand right SFOA includes biomass burning (which is usually said to give BBOA), but also coal and charcoal.*

Response: We have now added the BBOA factor as well as the following sentence in the Introduction explaining the difference between SFOA and BBOA: "The SFOA factor is a more general version of BBOA as it includes (in addition to biomass) other sources such as coal and charcoal."

11. *P6, L173. It could be noted that the Jathar et al. (2014) study also suggested different ratios of IVOC to PM than those of Shrivastava et al.*

Response: The emissions of "unspeciated non-methane organic gases" in Jathar et al. (2014) are still based on the same measurements as is Shrivastava et al.'s estimate (although they also included a couple of newer studies and averaged the estimates, so the numbers are slightly different). We believe that our paragraph about Jathar et al. in the Introduction is sufficient and that there is no requirement to add more about this to the Methods.

*12. P7, L191. 'under modeled SOA' - do you mean when comparing with observations?*

Response: Yes, this was what we meant. We have now added "when comparing with observations" to the end of this sentence.

*13. P7, L200. Shouldn't you also mention aromatics and other compounds here.*

Response: In the Dunmore et al. (2015) paper, the authors describe a quantification technique which uses the grouping of similar species in a lumped analysis based on carbon number and functionality. Given the separation of VOCs in a two dimensional space (from the use of a comprehensive two dimensional gas chromatography system), the aliphatic and aromatic compounds could be quantified separately. In our analysis, we only include the additional aliphatic IVOC species observed as their dominant emission source is likely from the use of diesel engines. We have now added the word aliphatic to the following sentence in the section "Additional IVOCs from diesel": "In this study, aliphatic IVOC emissions from diesel vehicles were introduced into the model proportionally to on-road transport VOC emissions,…"

*14. P7, L207. Which GC x GC system? Explain what is meant.*

Response: We have now replaced "GC x GC" with "measured by a comprehensive two dimensional gas chromatography (GCxGC) system (Dunmore et al., 2015)."

*15. P7, L209. Any reference for the number of studies providing that rate constant?*

Response: We have added a citation to the review article by Atkinson and Arey, 2003.

*16. p8, L247. Biogenic emissions of what?*

Response: We have added the word "VOCs" so text now reads "biogenic emissions of VOCs".

*17. p9. Sect. 2.6 'Comparison with measurements': This section can be simply renamed 'Measurements, since that is what it deals with down to L288.*

Response: We would like to retain "comparison with measurements" in the section title since (i) the IVOC emissions were also based on measurements, but were not used to compare the model with, (ii) then we can keep the evaluation statistics in the same section, reducing the number of short sub-sections.

*18. P9, L259. Why have references to Fig. 3? Give the references after the mention of each site, or add 'site details given in' or some such phrase.*

Response: We have moved this sentence to after the citation of the references.

*19. P9, L274. Are you sure that European inventories don't include cooking OA? I think it may be underestimated, but am not sure it is ignored completely.*

Response: We are sure that COA is not included in the UK National emissions inventory (Tim Murrells 2016, personal contact; NAEI 2013) Fountoukis et al. 2016 also claim that COA is not included in the European inventory they use. We have changed the sentence in the manuscript to read: "As our emissions inventory does not include cooking OA (NAEI, 2013), this factor could not be compared with the model."

*20. P9, L276. This sentence was confusing. I can see that two instruments can disagree, but what does it mean if there is just one instrument? Can an AMS and its PMF disagree, or what?*

Response: We agree that this sentence was not entirely clear and have amended it to: "When AMS measurements and their PMF apportionments are compared, some disagreement is observed, as shown for the two instruments measuring at the same time at the same location at London North Kensington."

*21. P9, L284 .... what period/site/analysis are these sentences and statistics referring to?*

Response: We have added "at the London North Kensington site during the winter IOP" to this sentence.

*22. P9, L289 on. This small section on statistical metrics has nothing to do with the discussion of AMS etc which it follows, and could be set in a small section of its own.*

Response: This section is called "comparison with measurements" so we believe having the statistics here is appropriate. See also our response to comment no. 17.

*23. P9, L291. I don't think correlation coefficient needs a reference to Carslaw and Ropkins; 'r' has been used for many many years before that paper was written. Actually, NMGE might need more explanation. All these could usefully be defined in supplementary.*

Response: We have removed this citation and added the equation for NMGE. We would like to keep this section in the main paper as it is not taking up much space and we are trying to reduce the number of times the reader is referred to the supplement.

*24. P10, equation (1): explain what i and n are.*

Response: The necessary explanations have now been added.

*25. P10, L298. Re-phrase - it sounds as though the measurement mean is better at capturing the variation in measurements than the model.*

Response: Now re-phrased to read: "a zero or negative COE implies that the model cannot explain any of the variation in the observations".

*26. P10, L303. This bit about WRF could be moved to Sect. 2.1.*

Response: This text has been moved as suggested.

*27. P11, around L35. All these numbers for NMB, etc. could be tabulated for easier comparison.*

Response: These numbers are also given as labels in Fig. 7 in a consistent, comparable manner. We anticipate this will become clearer and easier for the reader when the paper is properly typeset and the figures located within the text, rather than at the end.

*28. P12, L364. This refers to Fig. 12a,b, but there are no a,b labels in Fig.12*

Response: There were small labels on Fig. 12 which we have now replaced with larger ones.

*29. P12, L389. Re-phrase (or omit). It is obvious from the plots that this background OA is an overestimate for some days at least.*

Response: We agree; the sentence referred to has now been omitted.

*30. P13, L405 on. The whole discussion here is in terms of SOA and IVOC. But, how did the model perform for NOx and CO for these 'difficult' periods - maybe the problem is dispersion rather than IVOC? Or maybe the model's enthalpy values are wrong, and don't respond to cold temperatures as they should. I don't see why problems are blamed on domestic sources either. Wouldn't for example cold-starts for vehicles also produce more POA/IVOC, or commercial premises use more fuel in cold conditions? Are wood-burning emissions really an issue in London?*

Response: The model performance for NOx is already presented in the paper (Figure 6a) and the performance for NOx during this period is much better than that of SFOA. We believe that domestic emissions are the most likely culprit here as local emissions of wood-burning are indeed an issue in London. We have added the following sentences: "Furthermore, London is a *smoke control area* and therefore no residential emissions of SFOA are assumed by the national emissions inventory for this area. However, recent studies have suggested that there are indeed local sources of SFOA in London (Crilley et al. 2015, Young et al. 2015)."

*31. P15, L475. The title says comparison to previous (IVOC=1.5xPOA) approach, but as noted above, the method used here seems to be unique; not that of earlier papers.*

Response: We have provided detailed response to this comment where it was first made above (1st major comment). In brief, we have undertaken new model simulation for the 1.5xPOA experiment and have amended the text accordingly.

*32. P15-P16. The authors make various policy recommendations, e.g. (P15, L500) 'refinements should be reported to CEIP' and a very specific recommendation for PM1, PM1-2.5 and PM2.5-10 on P16, L533. Why not just suggest submission of size distributions? Why no mention of volatility - the VBS approach almost begs for people to submit emissions in different volatility classes. And since this paper is really exploring IVOC and not PM emissions per se, why didnt the authors focus on those?*

*Actually, I suggest that the authors don't try to tell countries what to do, but rather discuss any scientific insights into emission reporting that this study on IVOC reveals.*

Response: We have now removed the two sentences that were directly referring to "what other countries should do". The suggestion for the incorporation of more particle size distribution data in inventories was already in the Discussion (a few paragraphs further down from the point in question). We have also now added the suggested recommendation of categorising PM emissions in terms of volatility classes (to emphasise a comment we made on this already in the Introduction):

"We showed that treating POA as semivolatile and letting it evaporate lead to a great underestimation of HOA and SFOA concentrations compared to measurements at the London North Kensington urban background site. As has been highlighted by a number studies before us (listed in the Introduction), we would also emphasize that a major source of uncertainty in OA modelling is the volatility of primary emissions, an issue that currently not addressed by official emissions inventories."

*33. P16, L509. Can't small changes in SOA (or any pollutant) also be a reflection of longrange transport? Not all pollution is formed at short time-scales close to source.*

Response: Yes, we agree and we now mention long-range transport in the sentence in question: "A relatively small daytime increase of SOA could be explained by the expansion of the boundary layer height (Xu et al. 2015), as well as by contributions from long-range transport."

*34. P16, L520. Where did the value 4.0e(-12) come from for ASOA and BSOA oxidation?*

Response: The value came from Lane et al. 2008; we cited this in the Methods section, but have now added the citation to the Discussion as well.

*35. P16, L520 on. This section offers a few 'tuning' suggestions, but there are always any number of these in the field of SOA formation. For example, recent studies have suggested that SOA formation should be much greater than previously assumed, perhaps by a factor of four or so (Zhang et al., 2014).*

Response: We agree; we have simply made some suggestions.

*36. P16, L527. I would say that this paper illustrates the potential for a significant contribution, rather than that they can quantify the relative impact. To do the latter, one would need to be sure that all relative impacts are reasonably well known, and that clearly isn't the case.*

Response: Whilst we agree with the general sentiment of this comment, our response to this question is the same as Major Point no.2: We derive the "30%" from the change in NMB (so in comparison to the measurements), not in comparison to total modelled SOA, and therefore we believe this is an appropriate number/value for our statement highlighting the potential contribution this addition has.

*37. P17, Sect.5. The first paragraph simply repeats sentences from Sect. Don't do that. This section should also mention the results for other pollutants (NOx, O3, etc.), which are the main reason one can have some confidence in the basic modeling system. (Use of such data is one of the strengths of this paper I think.)*

Response: The text has been reworded so that it is not a direct copy and the results for other pollutants are mentioned.

*38. P17, L563. imported from where?*

Response: We have modified the sentence to include "mainland Europe". It now reads: "…this was caused by an intense pollution plume with a strong gradient of SOA from mainland Europe passing over the rural location…"

*39. P17, last paragraph. The interpretation of what contributes to the 90-th percentile is not so easy I think. And I certainly don't think one can state that 40% is due to missing diesel precursors. SOA is too complex for such simple statements.*

Response: We agree that SOA is a complex issue for making delimitative quantitative statements; however, the modelling work points us in a certain direction. We have amended the sentence to state that the influence of missing diesel precursors is even greater on high percentile SOA days than its contribution to annual average SOA (removing the statement of "40%"). We have also removed the statement from the abstract. The sentence in the Conclusions now reads:

"Moreover, the 90-th percentile of modelled daily SOA concentrations for the whole year is 3.8 $\mu$g m$^{-3}$, and the influence of missing diesel precursors is even greater on high percentile SOA days that its contribution to annual average SOA. "

*40. P20, L635. Expand/explain CEIP. Is this the name of a report, or just a web side?*

Response: Expanded, see also the response to question no. 41.

*41. P21, L665, EEA, Entec - these references are too short for readers to understand or find. Give proper references, with addresses as necessary.*

Response: Thank you for pointing this out. Something happened to these references during the formatting of the manuscript (we did have proper URLs for these references at some point). We will make sure they are properly included in the revised manuscript.

*42. P29, Fig. 1. The caption should state early that this is PM25 \*and\* SVOC/IVOC gases. (The issue of IVOC or SVOC+IVOC is discussed above, but it complicates this figure.)*

Response: We think that mentioning IVOCs in the second sentence is appropriate, and early enough.

*43. P31. Fig. 5. Explain 'Base' as used in legend. P31. Fig. 6(a). NOx is a the sum of both NO and NO2. Were they really summed with own molecular weights, or is this as NO2?*

*(ppb would have been easier to interpret!). Also, in the captions, add the superscript ion-labels too.*

Response: Yes, the sum of NO and NO2 is represented as NO2. We have now added a note about this to the caption: "NOx (as NO2)"), as well as added the superscription labels.
We agree that ppb is easier to interpret for gases, but this plot also included particulates and we would like to use the same units for the different panels. Furthermore, European Air Quality Standards (http://ec.europa.eu/environment/air/quality/standards.htm) are also defined in mass units for gases.

*44. P33, Fig. 9. This Figure looks very fine on-screen, but when printing out it looks very different - much of the black seems to be over-written with green and/or blue. Please use a different figure format, and check the printout.*

Response: We thank the reviewer for this comment. We printed the manuscript out with a few different printers, and Fig. 9 looks fine with all of the printers we tried, but we did notice that some printers had problems printing our scatterplots. We have replaced the shading on Fig. 9 with lines and converted our scatterplots into a more printer friendly format/size.

*45. P34, Figs. 11-12. We don't really need these since Fig. 10 has made the point about gradients well. Move to Supplementary.*

Response: We agree that Fig. 10 is enough to make the point about gradient, but we also use Figs. 11 and 12 for showing how spatially variable SOA can be even on daily averaged maps (which include contributions from both import and from very local sources).

*46. P35, Fig. 14. Why use log-log plots? Wouldn't a simple liner plot better display the range of data?*

Response: We use log-log plots for 2 reasons: (i) it expands the lower range without losing information in the higher range (as on a linear scatter plot for a large dataset, the plotting symbols will overlap with each other much more in the lower range), (ii) log-log plots are have been used in other recent OA modelling studies (e.g. Fountoukis et al. 2014) making it easier to visually compare some of the results of model evaluation.

*47. P36, Fig. 16. Here the addDiesel statistics look worse than the base-results. Are these really consistent with data presented in Table 4?*

Response: These values are correct. The difference in NMGE, r and COE values is very small, so not really "worse". Note the (positive) change in NMB.

*48. Figs and color schemes. The colors used for ASOA, addDiesel, etc seem to vary randomly from figure to figure (e.g. green is addDiesel in Fig.9 but observed SOA in Fig. 17. Please harmonize.*

Response: Figure colours have been changed so that same colours are no longer used for different variables.

**References**

[revised manuscript text omitted]

---

## Author Comment (AC2) · 5 Apr 2016

**acp-2015-920: Simulating secondary organic aerosol from missing diesel-related intermediate-volatility organic compound emissions during the Clean Air for London (ClearfLo) campaign**

**Response to Reviewer 2 comments**

We thank this reviewer for their supportive and helpful comments. Below we respond to each comment individually.

The reviewer's comments are in italics and blue font, our responses in normal text.

*In this paper, Ots et al. present an interesting method to account for the emissions of intermediate volatile organic compounds (IVOCs). They suggest that VOC emissions can be added proportionally to VOC emissions as opposed to the POA emissions which is the standard method used by current Volatility Basis Set (VBS) models. This approach can potentially pave the way for an accurate representation of IVOCs in the emission inventories which is proved to be a necessity for SOA models during the last decade. Overall, the manuscript is well written and scientifically sound. I recommend this study for publication after taking the following comments into account.*
*General comment:*
>*The authors include additional diesel related IVOC emissions based on the VOC emissions from the transport sector. This resulted in a significant improvement of their model results which brought the predicted SOA close to measurements during winter, spring, and summer while it resulted in an overprediction during autumn (Fig. 17 of the manuscript). However, the transport sector consist only one of the ten sectors that their emission inventory includes. This raise the question of how much their model performance will change (towards overprediction) if they will add the missing IVOC emissions from the rest nine sectors. Do they have indications that the only important source of IVOCs is the transport sector? While I strongly support the suggested approach of deriving the IVOC emissions based on intermediate length alkanes (or naphthalene seen in other studies; Pye and Seinfeld, 2010) I am quite sceptic about the impact shown here by only one sector. I suggest adding a discussion on this matter, probably in section 4.*

Response: We thank the reviewer for this suggestion. The following text has now been added to the discussion:
"In our experiment of semivolatile POA (denoted 1.5volPOA), IVOCs were included from all source sectors. This experiment simulated substantially less ASOA than our addition of IVOCs associated with just the traffic source sector. This means that a combination of the POA-based and our addition of diesel-IVOCs proportionally to NMVOCs would not create a substantial overestimation of SOA concentrations compared to measurements. Nevertheless, further modelling studies (including different assumptions regarding ageing rates, fragmentation, and yields) as well as more measurements of IVOC emissions from different sources are clearly necessary."

*Specific comments*

*1. Page 2 line 20: Biomass burning OA (BBOA) is also a usual component that PMF can identify. Does SFOA correspond to BBOA? If so, you should use the latter since it is more commonly used by the AMS community. Furthermore, you should also report the oxygenated organic aerosol (OOA) which then can be split into LV-OOA and SV-OOA.*

Response: We have added OOA and BBOA to the list of PMF factors. We have also added a sentence explaining SFOA and BBOA: "The SFOA factor is a more general version of BBOA as it includes in addition to biomass other sources such as coal and charcoal."

*2. Page 2 lines 22-27: Since the main focus of the manuscript is the simulation of SOA, it would be good to add a sentence regarding the performance of the global models in terms of SOA (e.g.,Spracklen et al., 2011;Jathar et al., 2011;Jo et al., 2013;Mahmud and Barsanti, 2013;Shrivastava et al., 2015;Tsimpidi et al., 2016)*

Response: We have added the sentence:
"Global modelling studies of SOA specifically have demonstrated huge uncertainties (up to tenfold) in total simulated SOA budgets (Spracklen et al., 2011; Jathar et al., 2011)."

*3. Page 3 line 20: Please add recently developed models that follow the same assumption in order to indicate that the factor of 1.5 is widely used up to date (e.g.,Koo et al., 2014;Tsimpidi et al., 2014)*

Response: Thank you for these suggested additional references which have now been included. (Note that LatexDiff doesn't seem to handle differences in a long list of citations very well, so the line shoots off the paper border, but we assure you that all of the original references as well as the additional ones are in the revised manuscript.)

*4. Page 4: The line numbers here and in a number of the following pages are not correct. They should either restart in each page or continue throughout the text.*

Response: We believe this bug has been fixed in the latest Copernicus LaTex package (for the submission we used 4.0 updated on 14-Dec-2015, but the latest is 4.2 updated on 22-Jan-2016).

*5. Page 5 line 13: Do HOA and SFOA correspond to the fossil fuel combustion and domestic combustion of you emission inventory (EI)? Please clarify since it is not clear if you used these fractions to convert the OC from your EI to OA.*

Response: The splits applied to the national PM inventory included the total OM (HOA and SFOA) so we did not apply an additional conversion. We now realise how having this sentence right after the emission fractions is confusing, so we have moved this sentence into the next section (following the initial OM/OC ratios for the VBS species).

*6. Pages 5 line 28: Assuming that POA is treated as non-volatile will result in unrealistically high OA concentrations in the aerosol phase. This will favor the partitioning of your semivolatile compounds (e.g. oxidation products of IVOCs) into the aerosol phase resulting in an overestimation of SOA as well. On the contrary, if you do*

*not assume that POA and SOA participate in the same solution during the phase partitioning, you should expect an underestimation of SOA. Please comment at this point on the implications of your assumption regarding the POA volatility.*

Response: In our simulations, POA and SOA do participate in the same solution during the phase partitioning, but the over- or underestimation of POA does not have a significant effect on the absorptive partitioning of SOA (Figure A; the POA units are in µg m$^{-3}$, note the nonlinear scale of the colours, Part. means particulate – i.e. the amount that is in condensed phase).

[Figure]

Figure A. Annual average concentrations of SVOCs and IVOCs in the volatility bins modelled with the addDiesel experiment at the North Kensington measurements site with their gas-particle partitions coloured yellow or orange-red, respectively (at an ambient temperature of 10 °C).

*7. Page 6 line 2: Tsimpidi et al. (2010) used 4 volatility bins to distribute the oxidation products of VOCs. Can you please report the aerosol yields for the 5th volatility bin that you are using (C\*=0.1) and add a reference for them as well?*

Response: There are no initial yields for this bin from the VOCs, but VBS species will move into that bin via ageing. We have now changed the first sentence of this paragraph, as well as added a note about the lowest bin, to read: "Five volatility bins (C\* = 0.1, 1, 10, 100, 1000 µg m$^{-3}$) are used for SOA production and ageing. The SOA yields for alkanes, alkenes, aromatics, isoprene and terpenes under high and low NOx conditions were taken from Tsimpidi et al. (2010). Note that Tsimpidi et al. (2010) reported yields for the four VBS bins between 1 and 1000 µg m$^{-3}$. In this work, the lowest VBS bin (0.1 µg m$^{-3}$) is used for the ageing reactions, as well as for SOA from the additional diesel IVOCs (explained in the next section)."

*8. Page 6 line 6: According to Lane et al. (2008) the use of aging reactions improved their results compared to measurements from urban areas but resulted in a strong overprediction over rural areas. They attributed this discrepancy to a potential balancing of decomposition to smaller more volatile products (fragmentation) and production of more substituted less volatile products (functionalization) during the*

*photochemical aging of biogenic SOA. This was also confirmed by laboratory studies (Ng et al., 2006). Therefore they suggested that no ageing of biogenic SOA should be considered. Furthermore, the use of an ageing rate constant of 4.0 x 10-12 cm3 molecule-1 s-1 is kind of conservative compared to what is used lately by most models (i.e. 1.0 x 10-11 cm3 molecule-1 s-1; Fountoukis et al. 2014). According to the above, I suggest either changing your scenarios by using 1.0 x 10-11 cm3 molecule-1 s-1 as a rate constant and assume no ageing of biogenic compounds or to make a sensitivity test and investigate the effect of these assumptions on your results (especially regarding the rate constant).*

Response: We address these issues of uncertainty with ageing rates in the Discussion:

"We use an ageing rate of $4.0 \times 10^{-12}$ $cm^3$ $molecule^{-1}$ $s^{-1}$ for both ASOA and BSOA (Lane et al., 2008). This is slower than has been used in some other studies (for example, Tsimpidi et al. (2010) uses $4.0 \times 10^{-11}$ $cm^3$ $molecule^{-1}$ $s^{-1}$ : 10 times faster, or Fountoukis et al. (2011) uses $1.0 \times 10^{-11}$ $cm^3$ $molecule^{-1}$ $s^{-1}$: 2.5 times faster). A combination of lower initial SOA yields, but slightly higher ageing rates could possibly flatten the diurnal cycle of our modelled SOA, matching the measurements better. Therefore, an improvement for the detailed, hourly, evolution could be achieved by a sensitivity study of these yields and ageing rates. This does not, however, change the main scope and results of this paper which illustrate the relative impact of the diesel-IVOCs on SOA formation."

*9. Page 6 line 10: What is the saturation concentration of this "background OA" compound? Is this considered nonvolatile since it is very aged and highly oxygenated? Furthermore, please provide a reference for assigning a value of 0.4 µg m-3 to this compound. Does this value based on measurements?*

Response: Yes, the background OA is considered nonvolatile since it is very aged and highly oxygenated. We have now added this information to the sentence where it is first mentioned. We have also added a reference to Bergström et al. 2014, where this value is set based on measurements at background sites. The relevant text now reads: "A constant background OA of 0.4 $µgm^{-3}$ is used to represent the contribution of OA sources not explicitly included in the model (e.g., oceanic sources or spores; Bergström et al. (2014)). This background OA is assumed to be highly oxygenated and is therefore included under modelled SOA when comparing with observations (with an OM/OC ratio of 2.0 it is also assumed to be nonvolatile)."

*10. Page 7 line 35 (on the top of the page): What is the SOA mass yield that you used for the 1000 µg m-3 volatility bin? Please provide a reference as well.*

Response: Presto et al. 2010 did not report yields from this reaction to the 1000 µg $m^{-3}$ volatility bin. We have now added a note about this: "For the oxidation products of $C_{15}H_{32}$, SOA mass yields were taken from Presto et al. (2010): 0.044, 0.071, 0.41, 0.30 for the 0.1, 1, 10, 100 µg $m^{-3}$ bins, respectively (Presto et al. (2010) did not report a yield for the 1000 µg $m^{-3}$ bin)."

*11. Page 7 lines 28-29: A more fair comparison between the two approaches would be to add IVOCs proportionally to POA from sector 7 only as you did on your addDiesel scenario. Can you investigate this additional scenario as well?*

Response: Adding IVOCs from all sectors maximises the potential effect of this approach of adding more emissions. It also makes the experiment with that approach directly comparable to what has been done in previous studies: Shrivastava 2008 for example. Furthermore, the effect of even including all sectors for SOA in the 1.5volPOA addition is much lower than for our addDiesel experiment, thus doing a finer addition would not change our conclusion that diesel IVOCs are a much bigger source than previously thought.

*12. Page 9 Section 3.1: Since POA are assumed to be nonvolatile you would expect to overpredict their concentrations. Are the emissions so severely underestimated? Please report that the presented underprediction will be even more significant if you add the semivolatile character of POA.*

Response: See our substantive response to Reviewer 1 Major Comment 1 on this point, and for the statement of modifications made to Methods and Results Sect.3.6.

*13. Page 9 line 20: Please replace the "secondary pollutants" with "secondary inorganic pollutants"*

Response: Done.

*14. Page 10 lines 7-8: Add a reference to Fig. 7*

Response: Added.

*15. Page 13 line 30 (on the top of the page): How you calculated the 40%?*

Response: The value was calculated as follows:
relative difference = (addDiesel-Base)/addDiesel.
We have added the following extra explanation into the manuscript: "… is 40% (calculated as the difference between SOA modelled with addDiesel and Base, relative to addDiesel: (addDiesel-Base)/addDiesel)."

*16. Page 15 lines 4-5 (or 33-34): Pye and Seinfeld (2010) have used a naphthalene-like surrogate specie to describe IVOCs instead of the traditional "POA" method. Please refer to this work as well (maybe in the introduction).*

Response: We have added a reference to Pye and Seinfeld (2010) where we mention global SOA budgets, but it is our understanding that they used CO emissions to derive a spatial distribution for naphthalene emission and then scaled their naphtalene up to include all IVOCs (using similar values as the Shrivastava et al.'s 2008 POA based approach). It is not completely different to the POA based approach (but different enough not to be mentioned in our list of 1.5xPOA). We have changed the sentence "To our knowledge, this is the first study where

IVOC emissions are added proportionally to VOC emissions" to "This is one of the very few studies where IVOC emissions are added proportionally to VOC emissions."

**References**

Bergström, R., Hallquist, M., Simpson, D., Wildt, J., and Mentel, T. F.: Biotic stress: a significant contributor to organic aerosol in Europe?, Atmos. Chem. Phys., 14, 13 643–13 660, doi:10.5194/acp-14-13643-2014, http://www.atmos-chem-phys.net/14/13643/2014/, 2014.

Fountoukis, C., Megaritis, A. G., Skyllakou, K., Charalampidis, P. E., Pilinis, C., van der Gon, H., Crippa, M., Canonaco, F., Mohr, C., Prevot, A. S. H., Allan, J. D., Poulain, C4 L., Petaja, T., Tiitta, P., Carbone, S., Kiendler-Scharr, A., Nemitz, E., O'Dowd, C., Swietlicki, E., and Pandis, S. N.: Organic aerosol concentration and composition over Europe: insights from comparison of regional model predictions with aerosol mass spectrometer factor analysis, Atmospheric Chemistry and Physics, 14, 9061-9076, 10.5194/acp-14-9061-2014, 2014.

Jathar, S. H., Farina, S. C., Robinson, A. L., and Adams, P. J.: The influence of semivolatile and reactive primary emissions on the abundance and properties of global organic aerosol, Atmos. Chem. Phys., 11, 7727-7746, 2011.

Jo, D. S., Park, R. J., Kim, M. J., and Spracklen, D. V.: Effects of chemical aging on global secondary organic aerosol using the volatility basis set approach, Atmos. Environ., 81, 230-244, 2013.

Koo, B., Knipping, E., and Yarwood, G.: 1.5-Dimensional volatility basis set approach for modeling organic aerosol in CAMx and CMAQ, Atmospheric Environment, 95, 158- 164, 10.1016/j.atmosenv.2014.06.031, 2014.

Mahmud, A., and Barsanti, K.: Improving the representation of secondary organic aerosol (SOA) in the MOZART-4 global chemical transport model, Geoscientific Model Development, 6, 961-980, 10.5194/gmd-6-961-2013, 2013.

Ng, N. L., Kroll, J. H., Keywood, M. D., Bahreini, R., Varutbangkul, V., Flagan, R. C., Seinfeld, J. H., Lee, A., and Goldstein, A. H.: Contribution of first- versus secondgeneration products to secondary organic aerosols formed in the oxidation of biogenic hydrocarbons, Environ. Sci. Technol., 40, 2283-2297, 2006.

Pye, H. O. T., and Seinfeld, J. H.: A global perspective on aerosol from low-volatility organic compounds, Atmos. Chem. Phys., 10, 4377-4401, 2010.

Shrivastava, M., Easter, R. C., Liu, X., Zelenyuk, A., Singh, B., Zhang, K., Ma, P.- L., Chand, D., Ghan, S., Jimenez, J. L., Zhang, Q., Fast, J., Rasch, P. J., and Tiitta, P.: Global transformation and fate of SOA: Implications of low-volatility SOA and gasC5 phase fragmentation reactions, Journal of Geophysical Research-Atmospheres, 120, 4169-4195, 10.1002/2014jd022563, 2015.

Spracklen, D. V., Jimenez, J. L., Carslaw, K. S., Worsnop, D. R., Evans, M. J., Mann, G. W., Zhang, Q., Canagaratna, M. R., Allan, J., Coe, H., McFiggans, G., Rap, A., and Forster, P.: Aerosol mass spectrometer constraint on the global secondary organic aerosol budget, Atmospheric Chemistry and Physics, 11, 12109-12136, 10.5194/acp- 11-12109-2011, 2011.

Tsimpidi, A. P., Karydis, V. A., Pozzer, A., Pandis, S. N., and Lelieveld, J.: ORACLE (v1.0): module to simulate the organic aerosol composition and evolution in the atmosphere, Geoscientific Model Development, 7, 3153-3172, 10.5194/gmd-7-3153-2014, 2014.

Tsimpidi, A. P., Karydis, V. A., Pandis, S. N., and Lelieveld, J.: Global combustion sources of organic aerosols: Model comparison with 84 AMS factor analysis data sets, Atmos. Chem. Phys. Discuss., 2016, 1-51, 10.5194/acp-2015-989, 2016.

---

## Author Response (AR2)

**acp-2015-920: Simulating secondary organic aerosol from missing diesel-related intermediate-volatility organic compound emissions during the Clean Air for London (ClearfLo) campaign**

**Response to editor final comments**

Thank you for the acceptance of our paper subject to attention to a few minor revisions. We respond to these below (original comments in italics and blue font), and have uploaded the final modified manuscript.

*Thank you very much for the careful revision of your manuscript following the comments by the reviewers. There are a few points where I consider that more actions/clarifications are needed.*
*1- point 3 of reviewer 1 and point 9 of reviewer 2: background OA – you are considering as background OA the amount of aerosols not explicitly considered in the model and you are counting all of it in the natural SOA. I wonder whether a part of this 'background' aerosol is of anthropogenic origin and after transport in the atmosphere and atmospheric again become 'background aerosol'.*

Response: This may be the case. We have now added the following sentence after the point at the end of Section 3.4 where we present a value for the average (BSOA + Background OA)/ASOA ratio: "This value is based on the assignment of the constant background OA in the model to natural SOA, which is what it is intended to represent. Some of this may have some anthropogenic origin, and more research on the missing (or boundary condition) sources that this background constant represents is needed for accurate attribution of the biogenic vs anthropogenic relative contributions."

*2- point 6 of Reviewer 1 for section 2.1: please also refer to the chemistry and aerosol modules you are using.*

Response: The following sentence has been added to the relevant part of Section 2.1: "The chemical scheme used in this study is EMEP-EmChem09soa with the MARS equilibrium module for gas-aerosol partitioning of secondary inorganic aerosol (Binkowski and Shankar, 1995; Simpson et al., 2012)."

*3- Point 10 of reviewer 1: I suggest rephrasing the sentence: "The SFOA factor includes biomass aerosols that are the so called BBOA as well as OA from coal and charcoal combustion." Is this what you mean?*

Response: Yes, thank you. We have now used your rephrased sentence.

*4- Page 3, line 25: leave the reference in the text.*

Response: Yes, the sentence now reads: "Since then, several regional and global ACTM applications have adopted this factor of 1.5 (e.g., Murphy and Pandis (2009); Tsimpidi et al. 20 (2010); Hodzic et al. (2010); Jathar et al. (2011); Fountoukis et al. (2011); Genberg et al. (2011); Bergström et al. 2012); Zhang et al. (2013); Koo et al. (2014); Tsimpidi et al. (2016))."

*5. page 14, lines 20 and 21: remove parenthesis: Recently,....simulations.*

Response: Removed.

*6. page 16: in the conclusions you do not need to define again ACTM and IVOC.*

Response: Definitions removed.

[revised manuscript text omitted]

---

## Author Response (AR3)

Dear Editor,

Thank you very much for handling our paper. We have removed the words as requested:

days, the relative contribution to SOA from diesel IVOCs could be greater than 40% (calculated as the difference between SOA modelled with addDiesel and Base, relative to addDiesel: (addDiesel-Base)/addDiesel). We note that Fig. 17a shows that in the addDiesel simulation, the modelled BSOA+Background OA still makes up 53% of the SOA, as an annual average. This

10   value is based on the assignment of the constant background OA in the model to natural SOA, which is what it is intended to represent.  This may have some anthropogenic origin, and more research on the missing (or boundary condition) sources that this background constant represents is needed for accurate attribution of the biogenic vs anthropogenic relative contributions.

With best wishes,

Riinu Ots